# Bayesian model and selection signature analyses reveal risk factors for canine atopic dermatitis

Katarina Tengvall [1✉], Elisabeth Sundström [1], Chao Wang [1], Kerstin Bergvall[2], Ola Wallerman[1], Eric Pederson[1], Åsa Karlsson[1], Naomi D. Harvey[3], Sarah C. Blott[3], Natasha Olby [4], Thierry Olivry[5], Gustaf Brander [1,6], Jennifer R. S. Meadows [1], Petra Roosje[7], Tosso Leeb [8], Åke Hedhammar[2], Göran Andersson [9] & Kerstin Lindblad-Toh [1,6✉]

Canine atopic dermatitis is an inflammatory skin disease with clinical similarities to human atopic dermatitis. Several dog breeds are at increased risk for developing this disease but previous genetic associations are poorly defined. To identify additional genetic risk factors for canine atopic dermatitis, we here apply a Bayesian mixture model adapted for mapping complex traits and a cross-population extended haplotype test to search for disease-associated loci and selective sweeps in four dog breeds at risk for atopic dermatitis. We define 15 associated loci and eight candidate regions under selection by comparing cases with controls. One associated locus is syntenic to the major genetic risk locus (*Filaggrin* locus) in human atopic dermatitis. One selection signal in common type Labrador retriever cases positions across the *TBC1D1* gene (body weight) and one signal of selection in working type German shepherd controls overlaps the *LRP1B* gene (brain), near the *KYNU* gene (psoriasis). In conclusion, we identify candidate genes, including genes belonging to the same biological pathways across multiple loci, with potential relevance to the pathogenesis of canine atopic dermatitis. The results show genetic similarities between dog and human atopic dermatitis, and future across-species genetic comparisons are hereby further motivated.

[1] Science for Life Laboratory, Department of Medical Biochemistry and Microbiology, Uppsala University, Uppsala, Sweden. [2] Department of Clinical Sciences, Swedish University of Agricultural Sciences, Uppsala, Sweden. [3] School of Veterinary Medicine and Science, University of Nottingham, Sutton Bonington Campus, Leicestershire, UK. [4] Department of Clinical Sciences, North Carolina State University, Raleigh, NC, USA. [5] Department of Clinical Sciences, North Carolina State University College of Veterinary Medicine, Raleigh, NC, USA. [6] Broad Institute of MIT and Harvard, Cambridge, MA, USA. [7] Division of Clinical Dermatology, Department of Clinical Veterinary Medicine, Vetsuisse Faculty, University of Bern, Bern, Switzerland. [8] Institute of Genetics, Vetsuisse Faculty, University of Bern, Bern, Switzerland. [9] Department of Animal Breeding and Genetics, Swedish University of Agricultural Sciences, Uppsala, Sweden. ✉email: Katarina.Tengvall@imbim.uu.se; kersli@broadinstitute.org

Atopic dermatitis (AD) is a chronic inflammatory and pruritic skin disease with characteristic distribution of lesions triggered by allergic reactions involving IgE directed towards environmental allergens. The same clinical presentation can be seen in dogs with food-triggered AD and also in a subset of dogs having atopic-like dermatitis, in which IgE reactions cannot be detected. Canine AD most often has an early onset (before three years of age), a feature included in the diagnostic criteria[1]. AD in both humans and dogs has proved to be highly polygenic as well as with epigenetic and environmental risk factors involved[2,3]. The strongest and most extensively described genetic association in human AD is with the *Filaggrin* (*FLG*) gene located within the epidermal differentiation complex (EDC) gene region on human 1q21[4]. Many proteins crucial for epidermal differentiation are encoded by genes clustered in the EDC, which also represents an ultra-conserved micro-syntenic block in mammals[5]. Apart from *FLG*, many additional genes have been associated with human AD. A multi-ancestry genome-wide association study (GWAS) of 21 K AD cases and 95 K controls identified 31 loci (including ten novel)[6]. A recent genome-wide meta-analysis of AD (22 K cases and 780 K controls) reported 25 previously defined and five novel loci[7]. Both studies identified the strongest signal in the *FLG*-locus. Canine AD is overrepresented in certain dog breeds such as Golden retriever (GR), Labrador retriever (LR), German shepherd dog (GSD), and West Highland white terrier (WHWT)[8–11]. Multiple genetic loci have been reported from GWAS of canine AD in different breeds, *e.g.* GSD (chr27:19 Mb CanFam2.0)[12], WHWT (chr3:35 Mb CanFam3.1[13], and chr17:54 Mb CanFam2.0[14]), and GR (chr3:64 Mb CanFam2.0)[15]. In concordance with human AD genes, the associated regions in dog harbor genes implicated in both innate and adaptive immunity, inflammation, and skin barrier formation. However, replication and functional validation of these loci have been limited[16–18] and the genetic background in canine AD appears more complex than initially suggested, even within breeds[19].

A limitation in a traditional GWAS (e.g., linear mixed model, LMM) of a complex trait is that it primarily tries to capture a single or a few risk factors with high effect size when, instead, multiple risk factors with effects ranging from small to moderate are expected to jointly influence the development of a complex trait. A traditional GWAS tests each variant one at a time as fixed effects and does not account for linkage disequilibrium (LD) between variants. To account for multiple testing a stringent p-value is often used, which results in many false negatives and the variants declared significant may be overestimated. A Bayesian mixture model (BMM) estimates effect sizes of all variants simultaneously and treats them as random effects, thereby accounting for LD between variants. This results in fewer false negatives and also gives unbiased estimates of the larger variant effects[20]. The BMM has been adapted to genome-wide studies of complex traits, *e.g.*, in the application BayesR[20,21]. BayesR models the effects of variants using four normal distributions, including one with zero effect (assuming that the majority of variants has non-measurable effect on a complex trait) and up to 1% of the total genetic variance. BayesR performs better than other methods in finding true positives compared to the number of false positives[20]. The aim of the current study was to identify genetic risk factors for canine AD. Assuming that many risk factors with small-to-medium effects are involved in the disease pathogenesis, we applied the BMM BayesR methodology.

Dog breeds result from strong artificial selection of favored phenotypes. Homogeneity is further intensified by subsequent closing of stud books and within-breed selection is still ongoing where dogs with specific characteristics are favored. The resulting selective sweeps are visible as a decrease in haplotype diversity caused by the rapid increase in allele frequencies at loci controlling the traits under selection. The hitchhiking effect is the unintended increase in allele frequencies of nearby variants at loci controlling another trait or disease. A pleiotropic effect can also be expected when genes responsible for the desirable trait also affect other phenotypes. In small populations, such as dog breeds, drift can also result in a loss of genetic variation. In this study, we performed whole genome analysis for signatures of selection by using the cross-population extended haplotype test (XP-EHH)[22] to investigate if the selection for certain breed characteristics has also led to accumulation of risk variants for canine AD within any of the four studied breeds.

The purpose of the present study was to uncover the genetic complexity of canine AD in a novel manner to move beyond single-locus GWAS signals. We performed genetic mapping in four dog breeds predisposed to AD, using datasets consisting of samples from ~200–400 dogs per breed that were sampled in a joint international collection effort. We identify multiple disease risk loci and replicate, in the dog, the major genetic risk factor for human AD.

## Results

**Bayesian genome-wide association identifies fifteen AD-associated loci.** Following quality control (QC) and relatedness filtering, the final datasets used for analyses consisted of 321 LR (178 cases and 143 controls), 256 GR (143 cases and 113 controls), 219 GSD (106 cases and 113 controls), and 235 WHWT (137 cases and 98 controls) with imputed marker sets of ~400–600 K variants (Supplementary Tables 1 and 2 and Supplementary Fig. 1). Using BayesR, we identified a total of 15 AD-associated loci; 11 in LR, one each in GR and in GSD, and two in WHWT (Fig. 1a, b, e, f). Variants with absolute effect size ≥0.0001 were defined as effect variants and AD-associated loci were regions harboring effect variants at <1 Mb distance (Table 1 and Supplementary Data 1). In LR, the three associated loci harboring variants with the highest effect sizes were located on chromosome 34 (top effect variant was *ARL14* intronic), chromosome 4 (*ITGA1/ISL1* intergenic), and chromosome 36 (*UBE2E3/ITGA4* intergenic). One associated locus in GR was defined on chromosome 23 (*SCN5A* intronic), and in GSD on chromosome 9 (*ABCA9* intronic). The two loci in WHWT were on chromosomes 10 (*HMGA2/LLPH*, intergenic) and 15 (*C4orf45* intronic). The sum of risk alleles of the 11 loci (i.e., risk index) differed in cases compared to controls of LR (two-sided *t*-test $p = 1.52 \times 10^{-22}$, t-statistic = 10.6, $n = 321$ dogs; Fig. 2 and Supplementary Data 2), and the AD variance explained by the risk index was 26.4% in LR. When modeling each associated locus separately, the total variance explained by the risk loci was 32.8%, with the largest contribution by chromosome 3 (7.3%). Principal component (PC)1, which captures the first dimension in the relationship matrix (PCA plot, Supplementary Fig. 1), contributed 9.3% to the total AD variance in the risk index model and 5.2% when modeling loci separately in LR (Supplementary Table 3). Associated loci in GR and GSD explained 2.3% of the AD variance in each breed respectively, and the risk index for the two loci in WHWT explained 16.1% of the disease variance (Supplementary Table 4). Both GR and GSD had a high influence by PCs on AD variance (in total 18.7% and 9.7% by PC1-3, respectively) whereas the contribution was low in WHWT (3.5% by PC1-2).

**The canine AD-associated locus on chromosome 17 is syntenic with a human AD risk locus.** The canine AD-associated locus on chromosome 17 in LR consists of nine effect variants across ~2 Mb and are grouped in two clusters; one at chr17:57.6-58.2 Mb

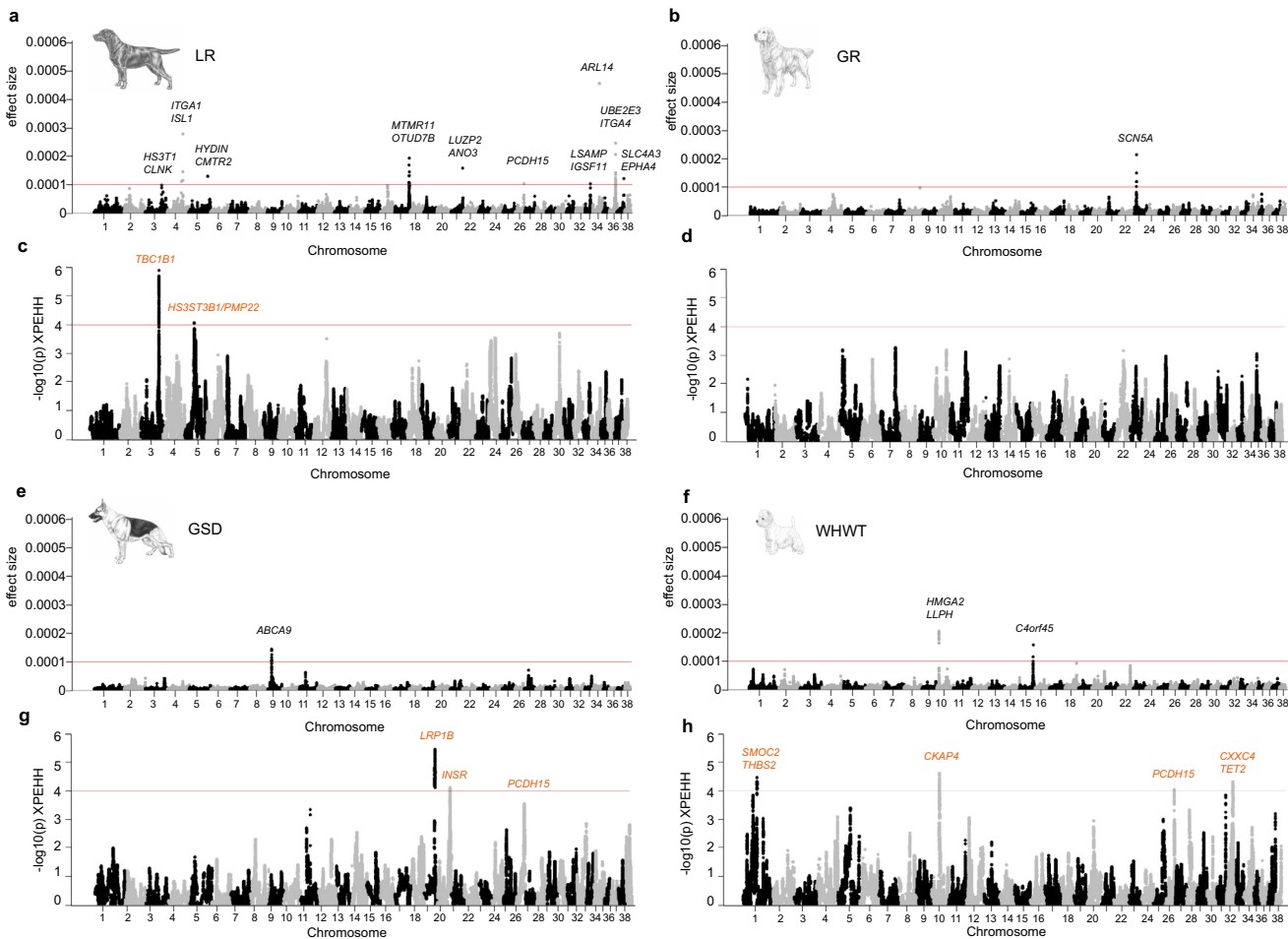

**Fig. 1 BayesR and XP-EHH regions associated with canine AD in four dog breeds.** The results from BayesR analyses are presented for LR (**a**), GR (**b**), GSD (**e**), and WHWT (**f**). The red line defines the cutoff for AD effect variants at absolute effect size of 0.0001 resulting in 11 AD-associated loci in LR, one locus in GR, one in GSD, and two in WHWT above this threshold. The closest protein-coding gene(s) to the top effect variants are specified for each locus. Panels **c**, **d** and **g**, **h** present genome-wide scans for signatures of selection in canine AD cases or controls for each breed and candidate regions under selection are defined by selection variants above the threshold line (orange; -log₁₀(p) XP-EHH = 4). Protein coding gene(s) closest to the top selection variants for each candidate region of selection are highlighted in orange.

(effect variants chr17:a-d) and one at chr17:59.1–59.7 Mb (effect variants chr17:e-i; Fig. 3). To search for additional candidate causative variants on chromosome 17, we performed long-read Oxford Nanopore Technologies (ONT) sequencing of four LR (two cases heterozygous for the risk alleles at the chromosome 17 effect variants and two controls homozygous for the non-risk alleles). Sequences from these individuals confirmed variants across the region; 486 of the called variants, extending >3 Mb (57.09–60.41 Mb), were in LD ($r^2 > 0.8$) with at least one effect variant in the whole LR dataset. The canine AD-associated locus in LR, extended with LD variants, ends ~0.5 Mb from the canine major EDC region located at chr17:61.0-62.0 Mb (Fig. 3b; lifted from human EDC coordinates[5]) and, according to the Broad Improved Canine Annotation v1 (canFam3.1), polyA transcripts in this region are primarily expressed in dog skin. Out of the 486 variants, 238 were heterozygous in cases and homozygous in controls of the ONT-sequenced dogs and of these 26 were located in canine ATAC-seq peaks from BarkBase[23]. Four of these variants were in both ATAC-seq peaks, ENCODE Candidate Cis-Regulatory Elements (cCREs)[24–26], and GeneHancer[27] elements (Supplementary Data 3). In addition, 133 novel variants were identified in the ONT-sequenced dogs with the same risk allele pattern and nine of these were located within canine ATAC-seq peaks out of which three overlapped both ATAC-seq peaks,

cCREs, and GeneHancer elements (Supplementary Data 4). We identified 65 structural variants (SVs) in the two ONT cases across the region chr17:55-65 Mb (Supplementary Data 5). Associated variants from two human GWASs overlapped with the LR risk locus; one associated variant from the human GWAS of AD[7] is located ~14 kb from effect variant chr17:c, upstream of *BCL9*, and one AD-associated variant from the human multi-ancestry meta-GWAS[6] is located in between effect variant chr17:h and chr17:i (Fig. 3). The region between variants chr17:a-d harbors the genes *FMO5*, *CHD1L*, and *BCL9*. By including a cluster of 30 variants (~58.2-58.5 Mb), in LD with chr17:d, the region was extended to contain the *ACP6* gene. A region of homozygosity (14 kb) spans half of *ACP6* and includes 19 variants in LD with chr17:d, out of which one variant overlapped both ATAC-seq, cCRE, and GeneHancer (Fig. 3d). There were 18 unique protein-coding genes within the region between the effect variants chr17:e-i. Variants chr17:e-h reside in the same canine topologically associating domain[28] (TAD) while chr17:i, intronic in the gene *ECM1*, resides in the adjoining TAD. From long read-based phasing, we concluded that one of the two cases presented a core haplotype (risk alleles following the same phased haplotype) between the two top effect variants chr17:f and chr17:g (Supplementary Table 5). Within the core haplotype, a 106 kb homozygous block (Supplementary Data 6) was identified in the

**Table 1 Canine AD-associated loci (BayesR).**

| Breed | Chr | N effect variants | Top effect variant | Position (canFam3.1) | Position (canFam4) | Nearest protein coding gene(s) | Genomic position | Effect size[b] | PhyloP | EA | OA | EA freq, cases (controls)[c] | VarExp (%) |
|---|---|---|---|---|---|---|---|---|---|---|---|---|---|
| LR | 3 | 1 | BICF2G630347384 | 68,495,098 | 69,059,690 | HS3ST1/CLNK, HS3ST1[a]/ RLOC_00020624[a] | Intergenic, intergenic[a] | 0.000104 | −0.01 | T | C | 0.070 (0.21) | 7.3 |
| LR | 4 | 1 | BICF2S23727494 | 58,135,165 | 58,993,320 | ANXA6 | Intronic | 0.000113 | −2.72 | T | C | 0.26 (0.38) | 3.8 |
| LR | 4 | 3 | 4,63451285 | 63,451,285 | 64,381,788 | ITGA1/ISL1 | Intergenic | 0.000280 | −0.02 | A | T | 0.17 (0.29) | 2.3 |
| LR | 5 | 1 | TIGRP2P74583 | 77,079,542 | 77,660,799 | HYDIN/CMTR2 | Intergenic | 0.000130 | −0.34 | G | A | 0.26 (0.34) | 1.7 |
| LR | 17 | 9 | 17_59225133 | 59,225,133 | 59,881,693 | MTMR11, OTUD7B[a]/ MTMR11[a] | Exonic, exonic[a] | 0.000194 | 3.94 | G | GACTC | 0.17 (0.28) | 1.7 |
| LR | 21 | 1 | 21_46702815 | 46,702,815 | 47,558,668 | LUZP2/ANO3 | Intergenic | −0.000159 | −0.34 | T | G | 0.37 (0.22) | 6.2 |
| LR | 26 | 1 | 26_34371008 | 34,371,008 | 34,592,644 | PCDH15 | Intronic | −0.000104 | 0.06 | T | C | 0.29 (0.16) | 3.8 |
| LR | 33 | 2 | TIGRP2P388667 | 21,007,172 | 21,647,630 | LSAMP/IGSF11, LSAMP[a] | Intergenic, intronic[a] | 0.000104 | −0.05 | C | T | 0.22 (0.37) | 3.7 |
| LR | 34 | 1 | 34_26660144 | 26,660,144 | 26,847,509 | ARL14, KPNA4/ ARL14[a] | Intronic, intergenic[a] | 0.000458 | 0.30 | G | A | 0.39 (0.48) | 0.3 |
| LR | 36 | 10 | 36_24333961 | 24,333,961 | 24,692,223 | UBE2E3/ITGA4 | Intergenic | 0.000247 | 0.30 | T | C | 0.077 (0.17) | 1.8 |
| LR | 37 | 1 | 37_27652421 | 27,652,421 | 27,537,344 | SLC4A3/EPHA4 | Intergenic | 0.000123 | −1.61 | G | T | 0.20 (0.25) | 0.2 |
| GR | 23 | 3 | 23_8319756 | 8,319,756 | 8,556,735 | SCN5A | Intronic | −0.000211 | −0.63 | A | G | 0.26 (0.34) | 2.3 |
| GSD | 9 | 8 | 9_15534865 | 15,534,865 | 11,391,956 | ABCA9, ABCA9[a]/ ABCA8[a] | Intronic, intronic[a] | −0.000143 | −0.35 | G | A | 0.16 (0.088) | 2.3 |
| WHWT | 10 | 9 | 10_8565303 | 8,565,303 | 8,921,678 | HMGA2/LLPH, CLO31:1[a] | Intergenic, intronic[a] | −0.000204 | 0.89 | T | G | 0.46 (0.27) | 7.3 |
| WHWT | 15 | 2 | 15_55988962 | 55,988,962 | 56,383,885 | C4orf45 | Intronic | 0.000156 | −0.07 | C | T | 0.31 (0.53) | 8.9 |

EA effect allele, OA other allele.
[a] canFam4 position (if different from canFam3.1).
[b] Effect size is negative if EA is the risk allele and positive if EA is the non-risk allele.
[c] EA frequency in the breed in which the effect was detected.

two controls; the two cases had in total 193 common heterozygous calls within the block and were thereby assigned haplotypes, whereas the controls only had one or two heterozygous variants throughout the region. The homozygous block spans the entire *VPS45* gene and 41 kb upstream towards start of transcription of *OTUD7B*, and one LD-variant, within the block, resided in an ATAC-seq peak represented in 16 datasets from various tissues and several individuals overlapped with a cCRE and a GeneHancer element (Fig. 3e and Supplementary Data 3).

**Selection analyses identify eight candidate regions.** The imputed datasets from 321 LR, 256 GR, 219 GSD, and 235 WHWT were also used in the XP-EHH analyses for detecting selection signatures in cases versus controls in each breed (Supplementary Tables 1-2). In LR, GSD, and WHWT, a total of eight candidate regions under selection (XP-EHH regions) were identified. Regions were defined using a 1 Mb window scan with 0.1 Mb overlap and at least two variants with -log₁₀(p) XP-EHH above 4 (Fig. 1, Table 2, and Supplementary Fig. 2). We investigated potential functionality of selection variants, i.e., variants with -$\log_{10}$(p) XP-EHH ≥4.0 ($N = 1471$), by extracting the phyloP[29] scores (Supplementary Data 7). We found that 12 selection variants were positioned at constraint sites (phyloP > 2.56; *i.e.*, showing a high level of conservation across 240 mammalian species and thereby a likely functional position) on chromosomes 3, 10, 19, and 32. One variant was exonic and the rest were intronic located within the genes nearest to the top selection variants per region (Table 2 and Supplementary Table 6). The variant with the highest phyloP score of 7.0 was exonic in *LRP1B* on chromosome 19 and is a missense variant XM_038565111.1:p.(Tyr42His; SnpEff v 4.3.t[30]) that exists in multiple dog breeds[31]. The putative impact of the exonic variant in *LRP1B* predicted in SnpEff was moderate and SIFT[32,33] predicted the substitution at amino acid position 42 to be tolerated with a score of 0.22 (SIFT score ranges from 0 to 1 and the amino acid substitution is predicted as damaging if the score is ≤0.05, and tolerated if the score is >0.05).

**Selection signal in Labrador retriever cases targets the *TBC1D1* gene.** A population substructure was discernible in the relationship matrix of LR (Fig. 4a and Supplementary Fig. 1a) and by utilizing information from Swedish LR kennels, questionnaires from UK, and coat color information from all LR, we concluded that PC1 likely captured a breed type division caused by selection for a gundog versus a common type LR. Gundogs were more often found in the low PC1 cluster, subsequently referred to as the gundog type, and the cluster with high PC1 values was considered as the common type (Supplementary Fig. 3). The 115 selection variants on chromosome 3 were positioned across the *TBC1D1* gene and the top selection variant, chr3:74,218,744 (chr3:sel), was intronic to *TBC1D1* (Fig. 4b). The allele C at chr3:sel was more frequent in the common type (Fig. 4a). *TBC1D1* is known for its association with body weight in humans[34,35], pigs[36], mice[37], rabbits[38], and chickens[39]. A stockier body is typically observed in common type LR, whereas the gundog is generally thinner, as illustrated in Fig. 4a. From the extended haplotype homozygosity (EHH) plot, we observed a higher integrated EHH (iHH; corresponding to the average haplotype length) for allele C at chr3:sel in cases (618 kb) compared to controls (205 kb; Fig. 4d, e). Along the extended region, estimated from the EHH plot for allele C in cases (Fig. 4d), AD-associated variants were defined using plink association (chi-square allelic test) and logistic regression models (Fig. 4f). LD between the risk alleles at chr3:assocA (also a defined effect variant in BayesR of LR) through assocD and allele C at chr3:sel was pronounced and this haplotype had a frequency of

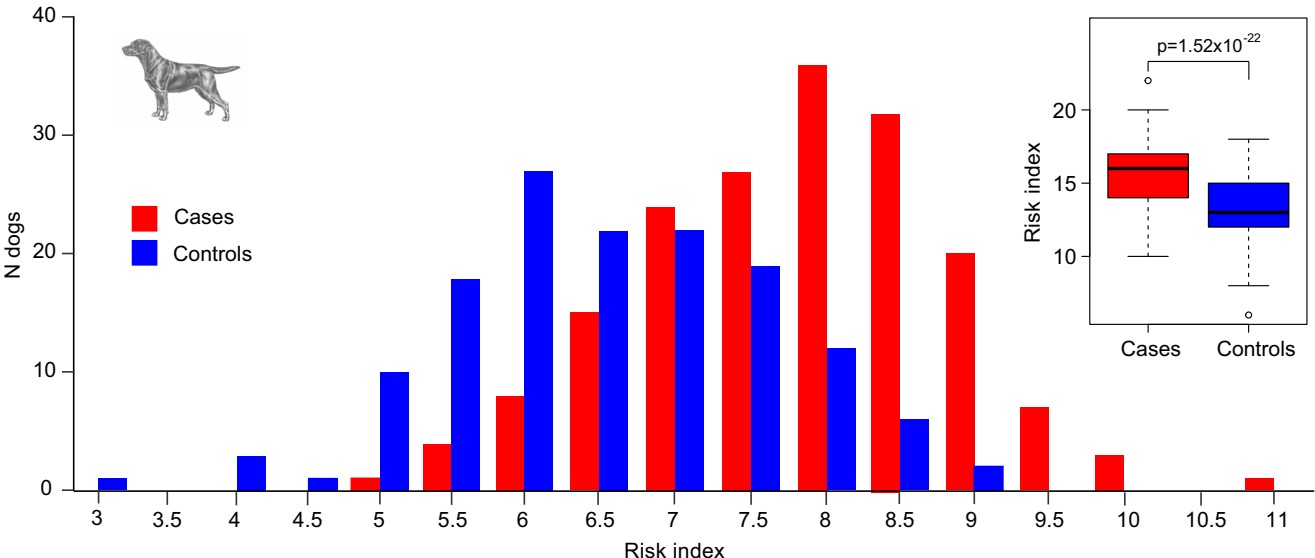

**Fig. 2 Canine AD risk index in Labrador retrievers.** In LR, the risk genotypes (no risk alleles = 0, one risk allele = 0.5, and two risk alleles = 1 per locus) at the top effect variant for each of the 11 AD-associated loci were combined into a risk index. The distributions of risk indexes were shifted with higher values in cases compared to controls and the mean risk index was significantly higher in cases (mean = 15.6) compared to controls (mean = 13.1; two-sided *t*-test $p = 1.52 \times 10^{-22}$, t-statistic = 10.6, n = 321 dogs). The difference is also visualized with boxplots, red (cases) and blue (controls), indicating median, first and third quartiles, and range of whiskers defined by max 1.5 of box length (Supplementary Data 2).

57.2% in the whole LR population, whereas the frequencies for the remaining nine haplotypes ranged from 1.0–9.9% (Fig. 4g). While selection is likely acting on the chr3:sel locus, the association with canine AD was stronger for the chr3:assocA-chr3:assocB-chr3:assocC risk haplotype CCG; chr3:sel genotype explains 25.1% of the PC1 variance and 4.4% of the AD variance, whereas CCG explains 18.4% of the PC1 variance and 7.6% of AD variance. The CCG frequency was 76% whereas frequencies for the other five haplotypes had a range of 1.5–8.8% (Fig. 4h). Among the 178 AD cases, 129 (72.5%) were homozygous CCG compared to 68 out of 143 (47.6%) controls (Fig. 4i and Supplementary Data 8). When dividing the dogs into subpopulations by setting the cutoff at PC1 = −0.05 (gundogs PC1 < −0.05 and common type PC1 > −0.05), it became clear that a large proportion of common type cases was homozygous CCG and that the CCG frequency was associated with AD in the common type ($\chi^2 = 17.5$, $p = 2.81 \times 10^{-5}$, $n = 245$ dogs; Fig. 6a and Supplementary Data 9).

**Selection signal in German shepherd controls across the *LRP1B* gene.** A division in the GSD breed into two subpopulations can be visualized in the PCA plot. We assigned the subpopulations to working type (PC1 < 0) and show type (PC1 > 0) based on the following information: GSD coming from kennels with a higher proportion of dogs with working merits compared to show merits were more common in the cluster with low PC1 values and vice versa, and GSD with black or gray coat color (typically observed among working type GSD) were almost exclusively present in the low PC1 cluster (Fig. 5a and Supplementary Fig. 4). A signal of selection consisting of 1078 selection variants was detected across the *LRP1B* gene on chromosome 19 in GSD (Fig. 5b). The top selection variant chr19:44,248,511 (chr19:sel) was located in the first intron of *LRP1B* (according to canFam4 and hg38) and a higher iHH was defined for allele T in controls (6.47 Mb) compared to cases (3.18 Mb; Fig. 5d, e). The association with AD was strongest around the *LRP1B* gene but in the logistic regression model, including covariates, the association was lost (Fig. 5f). The allele T at chr19:sel was more frequent in the working type

compared to the show type (Fig. 5a) and chr19:sel described ~15.2% of the PC1 variance, explaining the loss of AD association in the logistic regression model when correcting for PC1. The proportion of cases was higher in the show (63.6%) compared to the working type (33.0%), and the AD status explained ~12.5% of the PC1 variance indicating that the risk of AD differs between breed types of GSD, as suggested by us previously[12]. Homozygous T/T at chr19:sel was common among working type controls and the allele frequency at chr19:sel was associated with AD in the working type ($\chi^2 = 5.21$, $p = 0.0224$, $n = 107$ dogs; Fig. 6b and Supplementary Data 9).

The remaining XP-EHH regions were located on chromosome 5 (LR), chromosome 20 (GSD), and chromosomes 1, 10, 26, and 32 (WHWT; Supplementary Figs. 5–7).

**Genes in canine AD loci indicate joint pathways.** Using the UCSC browser (canFam4), we extracted 275 gene ID names and 268 transcripts with unassigned gene names in BayesR regions (±1Mb from effect variants), and 140 gene IDs and 130 transcripts with unassigned gene names in XP-EHH regions (Supplementary Data 10–11). Using *Homo sapiens* as the reference in STRING resulted in 193 recognized genes in BayesR regions and 136 genes in XP-EHH regions, whereas the reference *Canis lupus familiaris* resulted in 252 and 126 genes in BayesR and XP-EHH regions, respectively.

BayesR genes generated 20 significant terms (FDR < 0.05) in STRING (*Homo sapiens*) (Supplementary Data 12) with the most relevant term being from SMART: integrin alpha (beta-propellor repeats), including four *ITGA* genes (19 genes in the background count). *ITGA1* (chromosome 4) and *ITGA4* (chromosome 36) were located 995 kb and 320 kb away, respectively, from the top effect variants in the top two and three associated loci in LR. *ITGA10* positions within the ~2 Mb associated locus in LR on chromosome 17, and *ITGA9* is located ~500 kb from the top effect variant on chromosome 23 in GR where an *ITGA9* intronic variant had effect size 0.000099. BayesR genes in STRING (*Canis lupus familiaris*) resulted in no significant enrichments. Genes under putative selection in STRING (*Homo sapiens*) resulted in

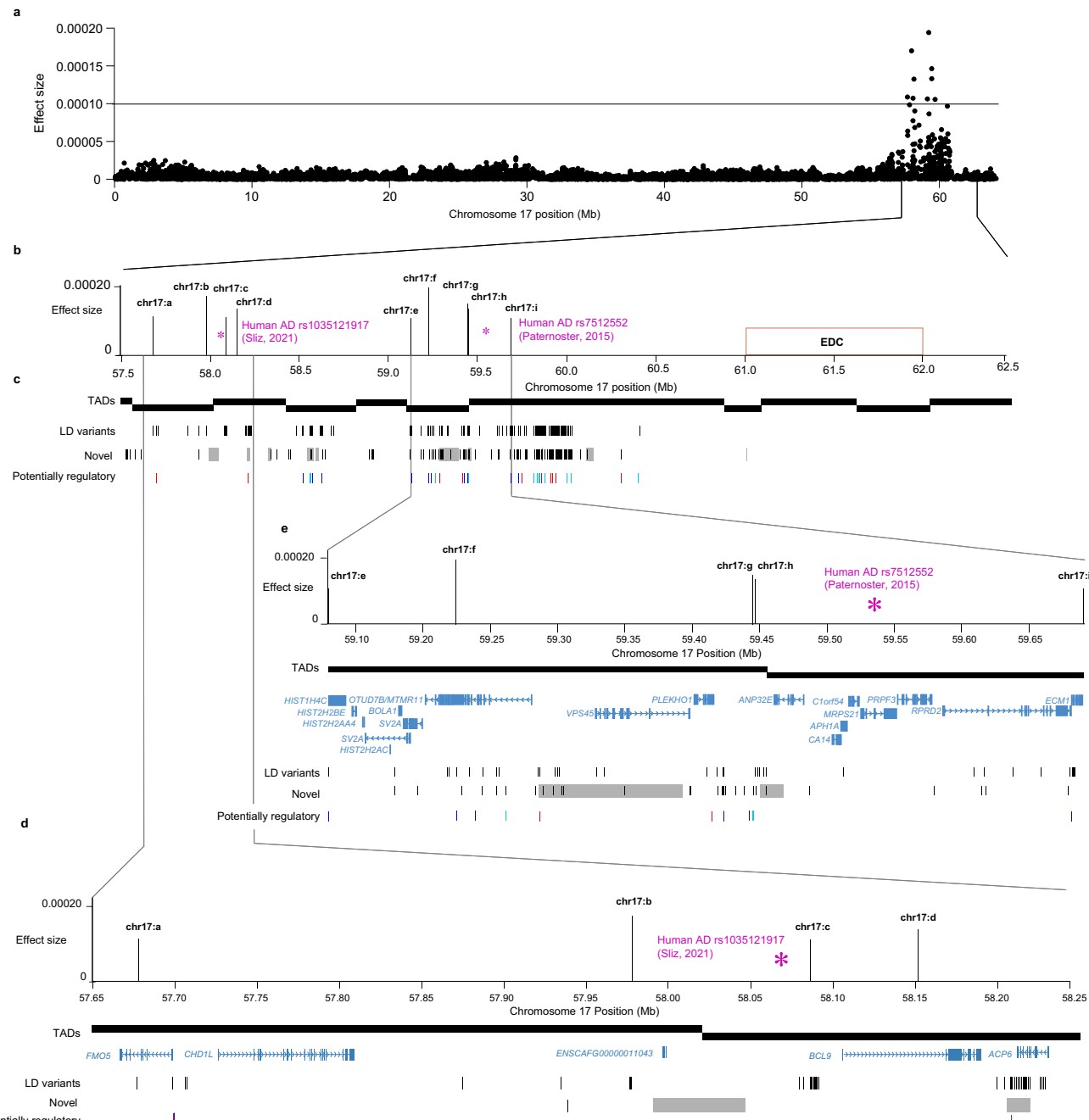

**Fig. 3 Canine AD-associated locus on chromosome 17 in Labrador retrievers.** One AD-associated locus in LR was located on chromosome 17 (**a**) with nine effect variants comprising a region of ~2 Mb. One of the top associated variants in human AD meta-GWAS[7] (Sliz et al., 2021, purple) was located ~14 kb from effect variant chr17:c and one AD-associated SNP from the human AD multi-ancestry meta-GWAS[6] (Paternoster et al., 2015, purple) was located in between variants chr17:h and chr17:i. Variants confirmed in sequenced dogs and in LD ($r^2 > 0.8$) with at least one of the effect variants extend >3 Mb (57.09–60.41 Mb), about 0.5 Mb from the epidermal differentiation complex (EDC) gene region (**b**). Black blocks show liver TADs in dogs. Black bars show 236 LD variants (plus two outside the figure boundaries at ~57.1–57.3 Mb) and 129 novel variants (plus four at ~57.3 Mb outside the figure boundaries). Variants with potentially regulatory functions are highlighted in light blue (overlap with canine ATAC-seq or both ATAC and cCRE), dark blue (ATAC and GeneHancer elements), and red (both ATAC, GeneHancer, and cCRE). Gray blocks show homozygous regions unique to the two ONT sequenced controls and not detected in the two sequenced cases (**c**). One homozygous region (56 kb) was found between effect variants chr17:b-c upstream of *BCL9*, and another region of homozygosity (14 kb) was spanning half of *ACP6* (**d**). One 106 kb homozygous region, including six LD-variants and five novel variants, overlapped the *VPS45* gene (**e**).

three terms related to leukemia (Supplementary Table 7). The genes in the leukemia cell line term were *AFDN* (alias *MLLT4*, chromosome 1, WHWT), *KLF3* (chromosome 3, LR), *RPS6*, (chromosome 5, LR), and *FCER2*, *MCOLN1*, and *PRAM1* on chromosome 20 (GSD). Genes under putative selection in STRING (*Canis lupus familiaris*) resulted in the significant GO Component term Phagocytic vesicle represented by the genes *APPL2* (chromosome 10, WHWT), *TLR1/TLR6* (chromosome 3, LR), and *RAB11A*, *RAB11B* and *STXBP2* on chromosome 20 (GSD). Additional significant terms were four STRING cluster

**Table 2 Candidate regions under selection (XP-EHH regions).**

| Breed | Region | N variants (% extreme) | N extreme (N phyloP>2.56) | Top variant ID | Position (canFam3.1) | Position (canFam4) | XPEHH | −Log10(p) | SA | OA | PhyloP | Nearest protein coding gene(s) | Genomic position |
|---|---|---|---|---|---|---|---|---|---|---|---|---|---|
| LR | chr3:73,100,000–75,200,000 | 909 (12.4) | 113 (3) | BICF2P1329770 | 74,218,744 | 74,791,636 | 4.84 | 5.89 | C | T | 0.17 | TBC1D1 | Intronic |
| LR | Chr5:37,200,000–39,100,000 | 538 (0.4) | 2 (0) | 5_3812310 2 | 38,123,102 | 38,340,254 | 3.92 | 4.05 | C | T | 0.10 | HS3ST3B1/PMP22, HS3ST3B1ᵃ/ ENSCAFG00000028937ᵃ | Intergenic, intergenicᵃ |
| GSD | Chr19:41,600,000–45,600,000 | 1868 (57.7) | 1078 (7) | 19_44248511 | 44,248,511 | 45,888,948 | −4.65 | 5.47 | T | C | −0.73 | LRP1B/ KYNU, LRP1Bᵃ | Intergenic, intronicᵃ |
| GSD | Chr20:51,100,000–53,100,000 | 718 (4.2) | 30 (0) | 20_52180278 | 52,180,278 | 52,588,044 | 3.97 | 4.14 | G | GTA | 0.10 | INSR/ ARGHEF18, INSR | Intergenic, intronicᵃ |
| WHWT | Chr1:55,100,000–57,100,000 | 832 (12.1) | 101 (0) | BICF2P921973 | 56,020,217 | 56,573,752 | −4.15 | 4.48 | C | T | −0.12 | SMOC2/THBS2 | Intergenic |
| WHWT | Chr10:30,900,000–33,400,000 | 863 (8.7) | 75 (1) | 10_32390517 | 32,390,517 | 33,443,853 | 4.23 | 4.62 | C | CCTGT | −0.36 | CKAP4 | Intronic |
| WHWT | Chr26:32,400,000–34,300,000 | 639 (0.6) | 4 (0) | 26_33382025 | 33,382,025 | 33,601,354 | −3.92 | 4.05 | A | G | −0.91 | ZWINT/PCDH15, PCDH15ᵃ | Intergenic, intronicᵃ |
| WHWT | Chr32:24,600,000–26,700,000 | 738 (9.1) | 67 (1) | 32_25613468 | 25,613,468 | 14,274,029 | 4.07 | 4.33 | TG | T | −3.53 | TET2/CXXC4, CXXC4-AS1ᵃ | Intergenic, intronicᵃ |

Column 2–4 refer to the XP-EHH region. Extreme variants have −log10(p) XP-EHH ≥4.
All columns following Top variant ID refer to the top variant.
SA selected allele, OA other allele.
ᵃcanFam4 genomic position (if different from canFam3.1).

terms, represented by genes from one or two regions only (Supplementary Table 8).

Combining genes from BayesR and XP-EHH regions in STRING (*Homo sapiens*) resulted in 23 significant terms (Supplementary Data 13) with the most relevant term from TISSUES: connective tissue represented by 37 genes (871 background genes). Seven BayesR regions (chromosomes 3, 4, 5, 10, 17, 23, and 34) and five XP-EHH regions (chromosomes 1, 5, 10, 20, and 32) were represented by the genes included in this network (Supplementary Table 9). Genes from BayesR and XP-EHH regions together in STRING (*Canis lupus familiaris*) resulted in the significant GO Process term: MyD88-dependent toll-like receptor signaling pathway (FDR = 0.004), represented by six genes (11 background genes). The genes from BayesR were *IRAK3* (chromosome 10, WHWT), *MYD88* (chromosome 23, GR) and *TNIP1* (chromosome 4, LR), and the genes *TLR1, 6* and *10* were from the XP-EHH region on chromosome 3 in LR. The other significant term was cellular anatomical entity (370 observed genes with 19,037 genes in the background).

In conclusion, canine AD candidate genes in BayesR and XP-EHH regions can be assigned to functions in the epidermis and/or immunity, and multiple genes in three BayesR and five XP-EHH regions were also detected in human GWAS of dermatitis, atopic eczema, eczema, and/or psoriasis (Table 3 and Supplementary Data 14–15).

## Discussion

We defined 15 AD-associated loci using BayesR, represented by 54 effect variants across four dog breeds. Our results present overlaps with human AD-associated regions and genes, which indicate that similarities exist also at the genetic level and not only in the clinical presentation and immunologic imbalance. The AD-associated locus on chromosome 17, identified with BayesR in LR, overlaps with associated variants from two human AD meta-GWAS studies[6,7] and harbor candidate genes from studies of mastocytosis[40] and eczema[41]. Several genes located close to the effect variants on chromosome 17 encode proteins that are relevant to skin and immunity. For example, BCL9 is a transcriptional coactivator associated with B-cell acute lymphoblastic leukemia[42], and is known to enhance transcriptional activity responses to Wnt signaling in both B- and T-cell lines[43]. Mutations in *VPS45* result in defective endosomal intracellular protein trafficking and severely defective neutrophils, which underlies an immunodeficiency syndrome in humans[44]. A neutrophilic skin infiltration is required for the development of chronic itch, and neutrophil depletion reduced itch-evoked scratching in a mouse model of AD[45]. Mutations in the *ECM1* gene cause lipoid proteinosis, a rare condition characterized by an abnormal skin thickening, suggesting that this protein is important for skin adhesion, epidermal differentiation, and wound healing[46]. ECM1 also has an important function in promoting M1 macrophage polarization, which is crucial for controlling inflammation and tissue repair in the intestine[47]. The top effect variant on chromosome 17 in LR resides in a potential regulatory region in between *MTMR11* and *OTUD7B*, overlapping 15 canine ATAC-seq peaks from nine different tissues as well as one GeneHancer promoter/enhancer element that is interacting with several genes in the TAD, *OTUD7B* being one of them. Another effect variant on chromosome 17, located 95 kb from the top variant and in the same TAD, also overlaps with a potential regulatory region covered by eight ATAC-seq peaks from four different tissues and one GeneHancer element in hg38. OTUD7B acts as a negative regulator of the non-canonical NF-kappa-B pathway and OTUD7B deficiency results in B-cell hyper-responsiveness to antigens[48]. It also plays a role in T cell homeostasis and normal T

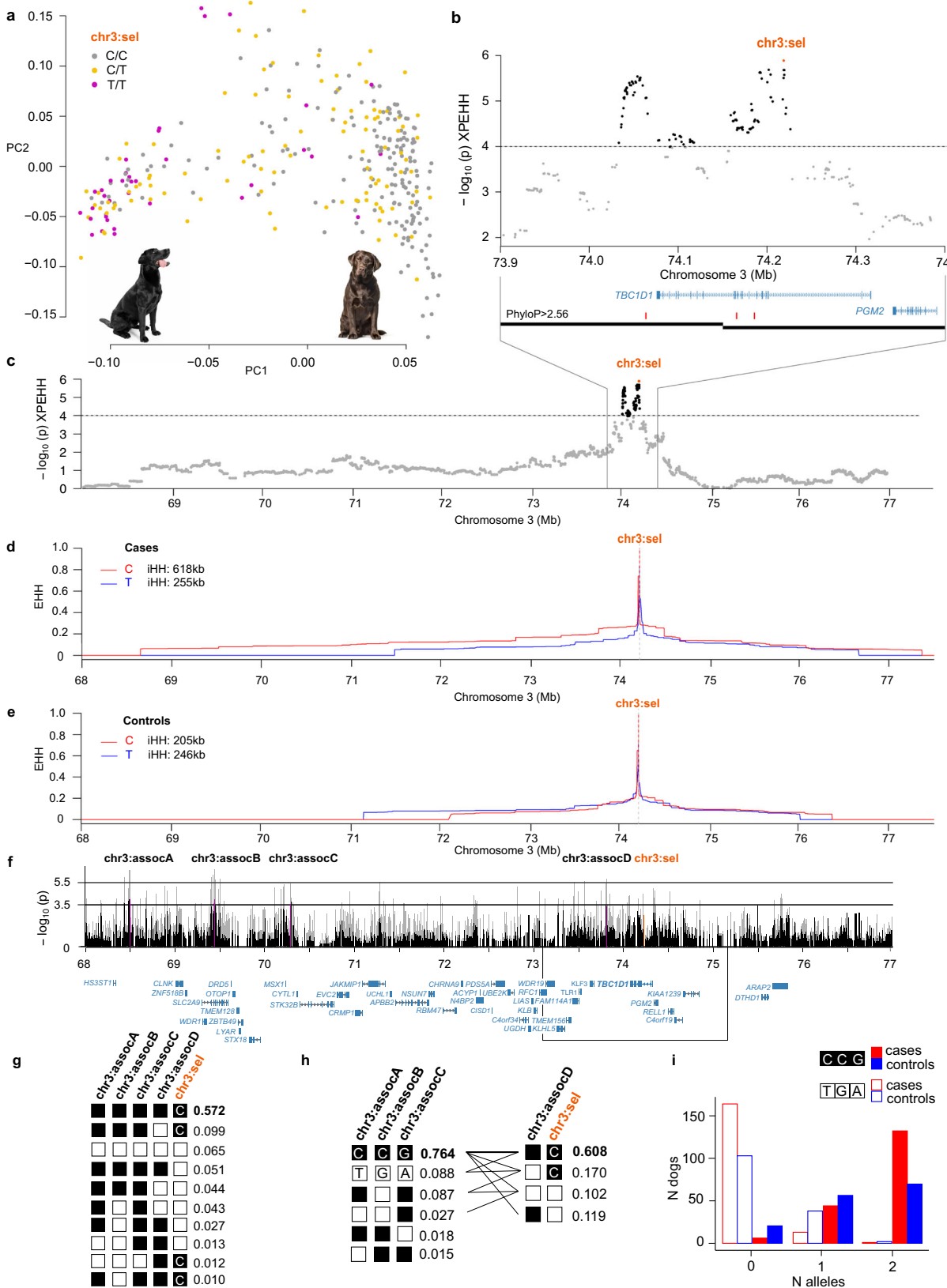

cell responses[49] and has been associated with eczema in human GWAS[41]. Several of the variants in LD with the effect variants on chromosome 17 also reside in canine ATAC-seq peaks, a few variants overlap peaks represented in more than 10 datasets, and the broadest overlap for a single variant is found in over 35 datasets representing all individuals and all tissues in the database

(interaction indicated with >17 genes in human). Some of these variants are also located within human ENCODE cCREs[24–26] and/or GeneHancer[27] elements, which indicates a potentially conserved regulatory function at these positions between dogs and humans. Also, the *PDE4DIP*[40], *OTUD7B*[41], *CIART*[6], *MRPS21*[6], and *SEMA6C*[7] genes in the canine AD-associated locus

**Fig. 4 A selection signal in Labrador retriever AD cases targets the *TBC1D1* gene.** The top XP-EHH region in LR was located on chromosome 3. Information about Swedish LR kennels, from UK owner questionnaires, and information regarding Switzerland working dogs suggest that LR of gundog type are present in the low PC1 cluster whereas the common type LR are represented in the high PC1 cluster. This is further supported by the chocolate coat color only found among the common type LR (Supplementary Fig. 3). The low PC1 cluster is subsequently referred to the gundog subpopulation, whereas the common type LR are in the high PC1 cluster. A heavier body type is typically observed in the common type, whereas the working type LR are generally thinner as illustrated by photos of typical gundog and common LR types (**a**). A signal of selection was detected around 74.0-74.3 Mb on chromosome 3 overlapping with the *TBC1D1* gene (associated with body weight). Two canine liver TADs (black bars) spanned the region. Red bars highlight variants with phyloP >2.56 (**b-c**). The allele C at the top variant position chr3:74,218,744 (chr3:sel) in LR was more frequent in the common type LR (**a**). A higher iHH in cases (618 kb) compared to controls (205 kb) was observed for the allele C at chr3:sel. The EHH plot of allele C indicates that LD decay is not complete until around 69 Mb and 77 Mb, respectively, in cases (**d**), and 72 Mb and 76 Mb in controls (**e**). Twelve genes were located within the candidate region under selection (black box). Association with canine AD within the extended region was calculated using plink --assoc (gray bars) and plink --logistic (covariates PC1 and PC2, black bars) resulting in associated variants along the haplotype; top variants (purple) named chr3:assocA through chr3:assocD (**f**). A high frequency (57.2%) was observed for the risk haplotype carrying risk alleles at chr3:assocA-chr3:assocD and allele C at chr3:sel, whereas nine different haplotypes (frequencies 1.0–9.9%) carried a mix of risk and non-risk alleles (**g**). LD was higher when considering only the chr3:assocA-chr3:assocC risk haplotype, CCG (frequency of 76.4%), compared to haplotype frequencies of 1.5–8.8% for the other combinations (**h**). Homozygous CCG was found in 129 (72.5%) of the cases and 68 (47.6%) of the controls (**i**; Supplementary Data 8).

on chromosome 17 were represented among associated genes from human GWAS of related diseases (Table 3 and Supplementary Data 14).

The highest effect size variant in LR was intergenic and positioned 44 kb from *ARL14* and 45 kb from *KPNA4* on chromosome 34. ARL14 controls the movement of MHC-II vesicles in human dendritic cells[50] whereas KPNA2 is involved in signal-transduction pathways that regulate epidermal proliferation and differentiation[51]. The effect variant on chromosome 37 in LR was intergenic between *SLC4A3* and *EPHA4*. EphA receptors and their ligands are expressed throughout all layers of the epidermis in human and in the basal layer of mouse epidermis, and are functionally integrated with intercellular adhesion complexes. Ephrin signaling complexes play a crucial role in epidermal cell–cell communication and regulate normal keratinocyte behavior. Alterations in the epidermal ephrin axis have been associated with wound healing defects and inflammatory skin conditions[52]. The *ANO3* gene on chromosome 21 (343 kb from the effect variant) in LR has been associated with eczema in humans[53].

One associated locus was defined in GR, consisting of three effect variants located within a ~17.5 kb region on chromosome 23. One effect variant (chr23: 8,186,340) resides in a region of canine open chromatin and overlaps with 16 ATAC-seq peaks in datasets from different tissues and individuals, as well as with a GeneHancer element when lifted to hg38. This GeneHancer element is assigned promoter and enhancer functions and interacts with several genes in the region. When lifted over to canFam4, this variant was located in the first exon of two longer *ACVR2B* transcripts. *ACVR2B* belongs to the type II activin receptor class and activin-A has been implicated in several aspects of immunity with fundamental roles in allergic responses and tissue remodeling in human allergic diseases, including allergic asthma and AD[54]. In mice, activin-A participates in the maintenance of skin homeostasis[55].

One associated locus on chromosome 9 was defined in GSD, with the top effect variant located in an intron of *ABCA9*. Several *ABC* genes, ATP-binding cassette (ABC) transporters, have been associated with skin disorders like *ABCA12* with keratosis pilaris[56] and Harlequin ichthyosis[57]. Furthermore, the transcriptional expression of *ABCA9* is induced during monocyte differentiation into macrophages[58] and macrophages are known to increase in numbers in acutely and chronically inflamed AD skin[59]. In concordance with the canine AD-associated *PKP2* (plakophilin 2)-locus previously described by us[12], the top 16th variant (effect size=6.8×10$^{-5}$, chr27:16,009,789) in the BayesR analysis of GSD was located 14 kb upstream of the *PKP2* gene.

One should note that the cutoff for defining effect variants is somewhat arbitrary and not strictly exact. In three breeds, only one or two associated loci were identified in the BayesR analysis, which indicates a too strict cut-off chosen because multiple risk factors were expected. However, lower effect sizes increase the risk of false positives[20]. In a study of anterior cruciate ligament rupture in LR, using BayesR, the top 50 effect variants were presented[60]. That approach in our study would result in additional markers in the already defined associated loci but also identify additional loci harboring variants with lower effect sizes, totaling in 50 associated loci (Supplementary Data 16). As a follow-up study, these loci could be further investigated or increased sample sizes could define additional lower effect variants of relevance to canine AD with higher certainty. A lower mean absolute effect size for each variant is also expected in a denser marker set because BayesR iterates the process of assigning variants to different effect size distributions and variants in high LD are randomly selected. On the other hand, the genomic positions indicated by effect variants are more precise and the risk of missing important regions associated with the trait is decreased in a denser marker set.

Artificial selection has caused a split in some dog breeds into morphologically and behaviorally divergent breed types. LR, GR, and GSD are examples of such breeds[61,62]. The original function of LR was to retrieve, but during the 1970s this breed became a popular pet dog resulting in divergence in selection goals establishing two types: one bred primarily for conformation or pet use (common type) and one for hunting (gundog)[61]. Differences in heritability of behavior traits have been found between these two LR types indicating variations in selection pressures[61]. A UK epidemiological study of health aspects in LR showed that overweight/obesity, ear (otitis externa) and joint conditions were the most common disorders affecting the breed[63]. Skin and ear diseases, including AD, were significantly more common among chocolate-colored compared to black- or yellow-colored LR[63,64] and chocolate-colored LR were heavier (on average 1.4 kg) than yellow or black dogs[65]. Genomic data has shown that the chocolate coat color is only represented in the genetic cluster of common type LR, whereas black or yellow were distributed across PC1[66]. Black LR had a higher fetching tendency than chocolate-colored LR, and chocolate-colored dogs demonstrated a lower trainability compared with black or yellow dogs[67], which indicates that gundogs rarely are chocolate-colored. Our results similarly point to a split between two LR types and we propose that the gene *TBC1D1* is the most likely target of selection in the common type LR. *TBC1D1* has been associated with body weight in multiple species[68], and it is located in the major QTL of growth

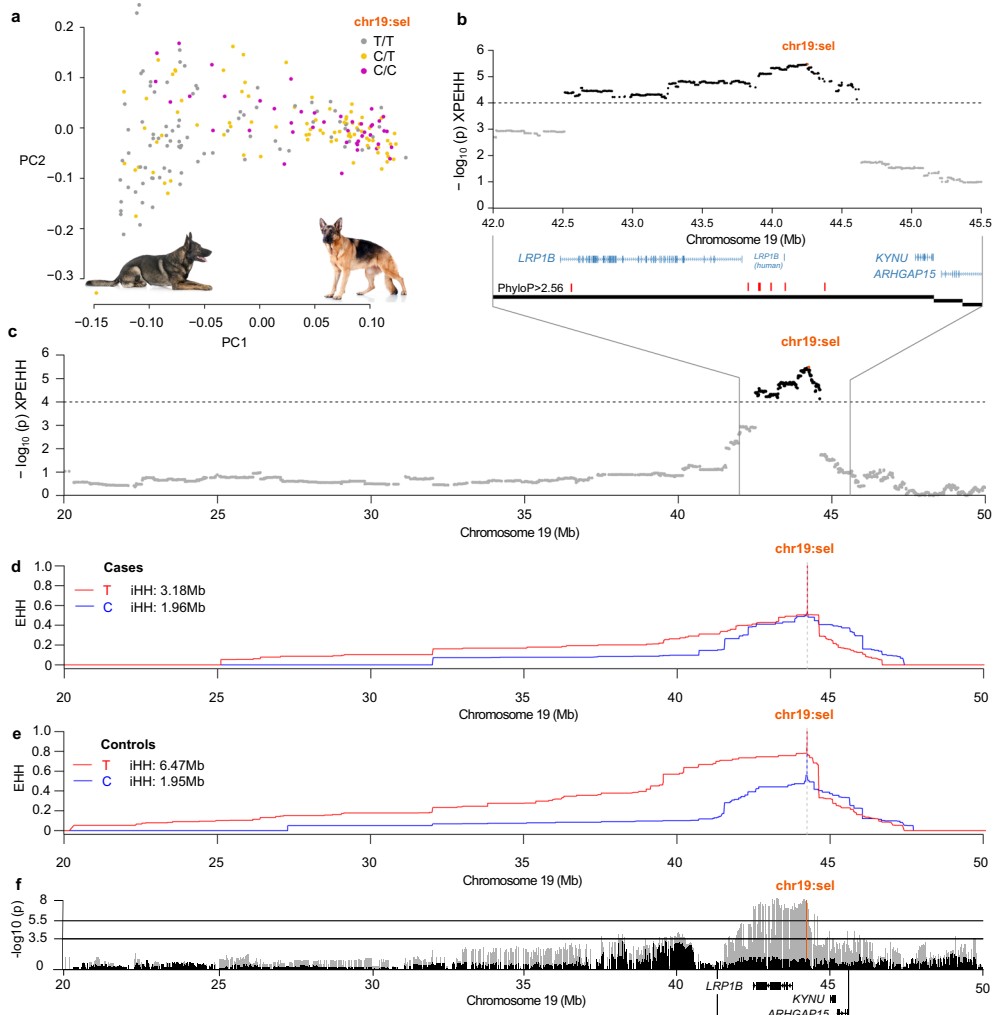

**Fig. 5 A selection signal in German shepherd controls spans the *LRP1B* gene.** Dogs from GSD kennels with a higher proportion of working compared to show merits and with gray or black coat color were more frequent in the low PC1 cluster whereas the opposite was seen for dogs in the high PC1 cluster (Supplementary Fig. 4). The subpopulations were subsequently referred to as the working (low PC1) and show (high PC1) type subpopulations. Dogs homozygous for the T allele were more frequent in the working type and the pictures illustrate the common gray color in a working type GSD and a show type with the typical GSD color pattern of brown with black saddle and a sloping cross (**a**). The selection signal on chr19 consisted of 1078 selection variants (-$\log_{10}$(p) XP-EHH ≥4) spanning the *LRP1B* gene and with the top selection variant located at chr19:44,248,511 (chr19:sel). The *KYNU* gene was positioned within the same canine liver TAD as *LRP1B* (black blocks). Red bars indicate selection variants with high phyloP scores (>2.56; **b**, **c**). The iHH for chr19:sel for allele T in cases (3.18 Mb) was lower compared with controls (6.47 Mb) indicating selection in controls (**d**, **e**). The association with canine AD (plink --assoc model, gray bars) was high across *LRP1B* and overlapping with chr19:sel. The signal was lost in the logistic model (black bars) where PC1, PC2, -$\log_{10}$(IgA), and -$\log_{10}$(Age) were included as covariates. Protein coding genes are presented within the XP-EHH region (black box; **f**; genes outside this region and three small transcripts without official gene symbols (*ENSCAFG00000024630, CFRNASEQ_PROT_00059550,* and *ENSCAFG00000030677*) within the region were excluded from the plot).

differences between broiler and layer chicken in three independent studies[69–71]. The extended haplotype that surrounds the selected variant in the cases, specifically, would appear to be the result of the original selection during breed type formation (possibly for a heavier body type given the known function of *TBC1D1*). This haplotype extends across the AD-associated variants on the left-hand side of the locus, and thus it can be assumed that they have increased in frequency due to hitch-hiking relative to the selected variant. The consequence of this is a pleiotropic haplotype that contributes to the selected phenotype, but also confers increased risk of disease. In the controls, however, the situation is different, as the selected haplotype now ends before the AD-associated variants, which can be explained by, at least, one recombination event separating the two parts and yielding a less genetically burdened version of the selected

haplotype. The canine AD risk haplotype CCG consists of the risk alleles at canine AD-associated variants positioned within or close to protein coding genes with possible connection to AD. For example, *CLNK* (506 kb from chr3:assocA/effect variant in LR and 367 kb from chr3:assocB) encodes a crucial signaling component of high-affinity IgE receptor induced mast cell degranulation[72]. WDR1 (98 kb from chr3:assocB) is involved in actin remodeling required when B cells respond to antigen-presenting cell (APC)-bound antigens[73]. The homeobox transcription factor MSX1 (53 kb from chr3:assocC), together with MSX2 and MOX1, is important for controlling dermal development, epithelial differentiation and proliferation into adulthood[74], and *KLF3* (142 kb from chr3:assocD and 555 kb from chr3:sel) encodes a transcription factor that controls gene expression during epidermal differentiation[75]. Also within the

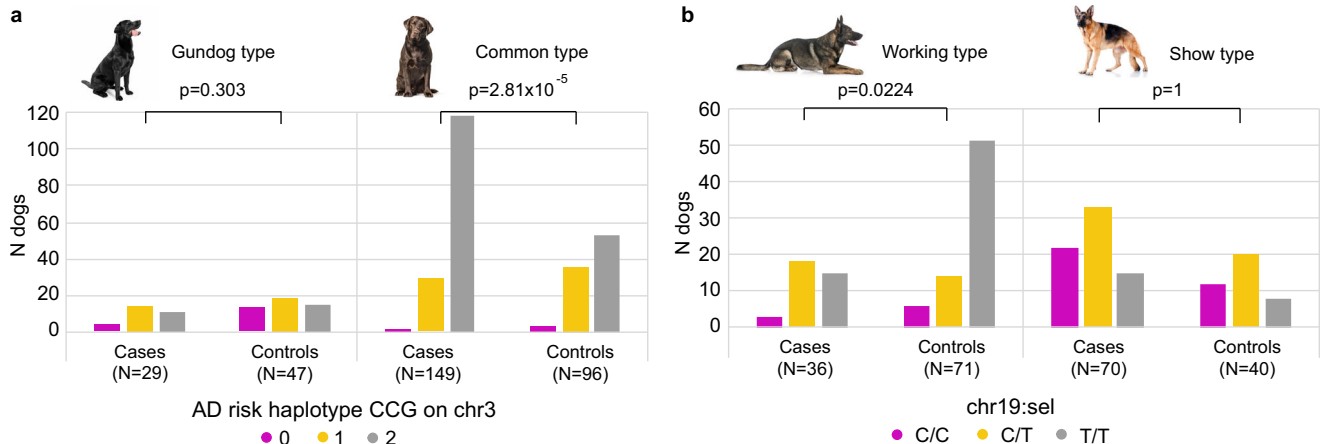

**Fig. 6 Allele frequency differences between breed types in both Labrador retrievers and German shepherds.** In LR, the number of cases with CCG (risk haplotype on chromosome 3) was significantly higher compared to controls of the common type ($\chi^2 = 17.5$, $p = 2.81 \times 10^{-5}$) whereas there was no significant difference in the gundog type ($\chi^2 = 1.06$, $p = 0.303$; homozygous = 2, heterozygous = 1, or homozygous non-risk = 0; **a**). In GSD, a homozygous T/T genotype at the top selection variant, chr19:sel, was more frequent among controls of working type ($\chi^2 = 5.21$, $p = 0.0224$) but in the show type there was no difference ($\chi^2 = 0$, $p = 1$; **b**). Dogs were split into subpopulations by a cutoff at PC1 of −0.05 in LR and 0 in GSD. Pearson's Chi-squared test with Yates' continuity correction of allele frequencies between cases and controls within each breed type was used. Two GSD controls from the working subpopulation did not have a genotype call and were excluded from the plot and this calculation (Supplementary Data 9).

**Table 3 AD-candidate genes in BayesR and XP-EHH regions with implicated functions in epidermis and/or immunity.**

| Breed | Chr | Epidermis | Immunity | Human GWAS (trait)[b] |
|---|---|---|---|---|
| *BayesR* | | | | |
| LR | 3 | – | *CLNK & WDR1* | *CLNK* (Vitiligo), *WDR1* (Psoriasis) |
| LR | 4 (~58 Mb) | – | *TNIP1* | *TNIP1* (Psoriasis) |
| LR | 4 (~63 Mb) | – | *ITGA1 & ISL1* | – |
| LR | 5 | – | – | – |
| LR | 17 | *ECM1* | *OTUD7B, VPS45, BCL9, & ITGA10* | *OTUD7B* (Eczema), *PDE4DIP* (Cutaneous mastocytosis), *CIART & MRPS21* (AD), *SEMA6C* (AD) |
| LR | 21 | *ANO3* | – | – |
| LR | 26 | *DKK1* | – | – |
| LR | 33 | – | – | – |
| LR | 34 | *KPNA2* | *ARL14* | – |
| LR | 36 | – | *ITGA4* | – |
| LR | 37 | *EPHA4* | – | – |
| GR | 23 | *ACVR2B* | *ITGA9, MYD88* | – |
| GSD | 9 | – | *ABCA9 (ABCA5,6,8,8B)* | – |
| WHWT | 10 | – | *IRAK3* | – |
| WHWT | 15 | – | – | – |
| *XP-EHH* | | | | |
| LR | 3 | *KLF3[a] & MSX1[a]* | *TLR1, 6, & 10, CLNK[a], WDR1[a]* | *TLR1* (AD), *TLR10* (Allergic disease) |
| LR | 5 | – | – | – |
| GSD | 19 | *KYNU* | *KYNU* | *LRP1B* (Asthma / Eczema), *ARHGAP15* (Eczema) |
| GSD | 20 | – | *FCER2* | – |
| WHWT | 1 | *SMOC2 & THBS2* | – | *SMOC2* (Vitiligo), *RIC8B* (Chronic inflammatory disease) |
| WHWT | 10 | – | – | – |
| WHWT | 26 | *DKK1* | – | – |
| WHWT | 32 | – | – | *INTS12 & GSTCD* (Psoriasis) |

[a]Genes in the extended haplotype region in Labrador on chromosome 3.
[b]References are presented in Supplementary Data 14–15.

defined XP-EHH region on chromosome 3 we find AD candidate genes; two *Toll-like receptor* (*TLR*) family genes, *TLR10* and *TLR1* (~640 kb from chr3:sel), which encode proteins that play important roles in the innate immune system. Several *TLR* genes have been associated with skin inflammatory diseases[76], including AD[77], and *TLR1* specifically has been linked to AD in children[78]. Both *TLR1* and *TLR10* have also been associated with AD and eczema in human GWAS[79,80] (Table 3, Fig. 4f, and Supplementary Data 15). The TLR-genes were also represented in the

MyD88-dependent TLR signaling pathway, defined with STRING, along with genes from three BayesR regions. This pathway includes activation of the NF-κB pathway, resulting in downstream activation of inflammatory cytokines.

Breed type specific selection in GSD was also evident in our data. The selection signal on chromosome 19 was detected in GSD controls of working type and extends across the gene *LRP1B*, which encodes a 4599 amino acid-long member of the low-density lipoprotein receptor (LDLR) protein family. *LRP1B* is

mainly expressed in brain and endocrine tissues in humans[81] and in the brain of mice[82]. According to the Broad Improved Canine Annotation v1 (canFam3.1), multiple transcripts were found in dog brain and kidney but not in other tissues. *LRPB1* has been primarily described as a cancer-driving gene[81], for example in multiple myeloma[83]. It has also been associated with asthma[84] and eczema[85] in human GWAS (Table 3 and Supplementary Data 15), and with cognitive decline[86], Alzheimer's disease[87], and infant cognitive ability[88]. The gene *KYNU*, encoding the enzyme kynureninase, is located 774 kb from chr19:sel and is the closest neighboring gene to *LRP1B*. *KYNU* presented with elevated expression in psoriatic skin lesions compared to normal skin in humans[89]. In normal skin, *KYNU* was primarily expressed in the basal layer of the epidermis whereas in the psoriatic skin, its expression was detected across the whole epidermis and in infiltrating immune cells (*e.g.*, T cells, macrophages, and dendritic cells)[89]. Induced psoriasis-like symptoms in mice were reduced after the application of *KYNU* inhibitors, and the knockdown of *KYNU* significantly inhibited the production of inflammatory cytokines in keratinocyte cell lines; altogether, these observations suggest that *KYNU* represents a likely therapeutic target in psoriasis[89]. Another neighboring gene, *ARHGAP15*, has been associated with eczema[41,80] but its functional implications are not directly related to the AD phenotype. Since allele T at chr19:sel was more frequent in controls of working type GSD, a possible scenario could be that specific variants affecting the *LRP1B* gene has influenced a work-desired trait in this GSD breed type that was selected for, and that additional (hitch-hiking) variants affecting (and potentially inhibiting) *KYNU* are AD-protective. Both alleles of the top three variants of LR and GSD XP-EHH regions on chromosomes 3 and 19 were present in wolves[90], thus these alleles are not unique to the breeds or to dogs in general.

Four integrin alpha genes located in different associated loci in LR and GR were highlighted in the STRING enrichment analysis. Integrins are heterodimeric transmembrane cell adhesion molecules with alpha (α) and beta (β) subunits combined in different dimers with diverse functions, for example in cell surface adhesion and signaling. Integrin alpha-4 subunit (ITGA4) associates with the beta-1 subunit in the integrin α4β1 in leukocytes, or with the beta-7 subunit in the integrin α4β7 present in a subset of memory T cells[91]. ITGA4 was upregulated in the non-lesional epidermis from horses suffering from insect bite hypersensitivity, an IgE-mediated dermatitis caused by insect bites and has common features with human and canine AD[92]. *ITGA9* mRNA expression was increased in human psoriatic skin[93] and overall, *ITGA*-genes, including the ones identified in the associated loci, have also been associated with human skin cutaneous melanoma[94]. The term connective tissue was represented by genes from seven BayesR and six XP-EHH regions, and, interestingly, a relationship between AD and autoimmune connective tissue disease including systemic lupus erythematosus, rheumatoid arthritis, and Sjögren's syndrome, has been described[95].

In conclusion, we detected multiple canine AD-associated loci, including one that overlaps with *FLG*, which is the major genetic risk factor described in human AD, and multiple candidate genes were assigned functions related to the epidermis and/or immunity and some were also detected in human GWAS of related diseases. We correlated within-breed selection with accumulation of AD risk or protective factors. The approaches used in this study have led us to better understand the complex genetics of canine AD in four dog breeds predisposed to this disease and implicate shared genetic causes between dog and human atopic dermatitis.

## Methods

**Phenotype classifications**. The clinical diagnosis of canine AD associated with IgE to environmental allergens was based on a set of stringent exclusion and inclusion criteria, including procedures in which solely food-induced AD cases were excluded to achieve a more homogenous group of likely environmentally allergy-associated AD cases with an established IgE-mediated response. In Swedish dogs, the diagnosis was made by first ruling out other causes of pruritus (e.g., ectoparasite infestation, staphylococcal pyoderma, and *Malassezia* dermatitis). Secondly, a hypoallergenic dietary trial (at least 8 weeks followed by a challenge period) to evaluate potential concurrent cutaneous adverse food reactions contributing to the clinical signs. Dogs not adequately controlled on hypoallergenic diets and with positive reactions on intradermal allergy tests or IgE serology tests were assigned the diagnosis of AD. Swedish controls were above five years of age without known skin disease or other immunological problems based on owner questionnaires and/or clinical examinations. American WHWT case and control phenotypic classifications were similar to the above description for Swedish dogs[13]. In the UK, LR and GR case and control phenotypic information was based on owner questionnaires[96]. The inclusion of AD cases from the UK was based on the following positive answers: (1) atopic dermatitis skin diagnosis (*N* = 286), (2) allergy tested (*N* = 161), (3) dietary trial and/or seasonal patterns (If no dietary trial was performed but seasonal pattern was defined, we required that the allergy test results defined IgE towards mites or other allergens). Dogs with negative (or inconclusive) results on allergen-specific IgE tests were excluded. In total, 67 LR and 57 GR cases respectively, were included with well-classified phenotypes from the UK sample set. The procedure for including UK controls was as follows: (1) including dogs with answer no on atopic dermatitis skin diagnosis (*N* = 295), (2) excluding dogs below five years of age, with food allergy, improving on exclusion diet, or had gastrointestinal problems, moist eczema skin lesions (hot spots), ear infection/inflammation, frequent vomiting, pruritus, or were allergy tested. In total, 48 LR and 68 GR were defined as well-classified controls for the current study. The majority of included cases and controls had available genotypes and were used in subsequent analyses. The phenotypic characterization of LR and GR cases and controls from Switzerland[97] was in line with criteria used for including Swedish cases and controls. An overview of sample distribution across countries is presented in Supplementary Table 1, including only the samples with genotype data available.

**Labrador retriever; breed types and coat color**. It is known that the LR breed has been split into a common type, bred for conformation and pet use, and a gundog type, bred for hunting[61,66]. We classified these breed types based on information from both Sweden, UK, and Switzerland. For the Swedish LR, we extracted the dog's kennel name and matched these with the LR breed club's criteria for LR kennels with breeding goals according to gundog focus. On the Swedish Kennel club (SKK) webpage all LR kennels in Sweden active since the 1970s are listed[98], and the Swedish LR breed club list gundog LR kennels (with puppies born since 2006)[99]. Of the 102 LR in our dataset with Swedish origin, 32 had kennel names included in the list of gundog kennels, 51 were not in this list, thus regarded as kennels of common type, and 19 had no specified kennel name or had a kennel name not listed by SKK. By extracting owner questionnaire information from UK LR, we identified seven gundogs, which clustered together with the Swedish gundogs (low PC1), and two show dogs clustering with the Swedish common type (high PC1). The Swiss LR cohort did not include any gundogs, but one police dog (low PC1, clustering with the gundogs) and 11 guide dogs for the blind (high PC1). Generally, the Swiss cohort clustered together with the Swedish/UK common type (high PC1) but also forming a subcluster partly overlapping with the common type, but not with the gundog subpopulation. The common type includes LR used for both dog shows, as pets, and for different kinds of work (guide dogs for the blind, snow avalanche rescue dogs etc. primarily represented by the Swiss LR). We also extracted coat color for the majority of the LR (all except 26) to evaluate if the chocolate coat color also supported the division into a common type versus gundog type of LR. Chocolate-colored LR were only present in the common type subpopulation (Supplementary Fig. 3) and represented 9.5% of the LR in the Swedish cohort, 20% in the Swiss, and 24% in the UK.

**German shepherd; breed types and coat color**. The GSD is also bred for either show or working capabilities, resulting in a split between two breed types. The number of working and show merits were extracted for in total 247 Swedish GSD kennels in total, with at least 50 registered offspring. Out of the 219 GSD included in our analysis, 121 had a kennel name, and of these, 30 were from kennels with a lower work proportion ($N_{\text{working merits}}/N_{\text{show merits}}$) <0.5, whereas 91 were from kennels with a higher work proportion ≥0.5, henceforth referred to as show and working type kennels respectively. We also extracted the registered coat color from SKK for 192 GSD. The different colors were subdivided into two color classes; (1) gray or black, including all dogs with the colors gray, dark gray, black with gray markings, and black, and (2) brown and black, which included dogs of black or gray color with brown, yellow or red markings. The most common colors were black with brown markings (*N* = 102), gray (*N* = 37), and black with yellow markings (*N* = 19). The remaining colors were assigned to seven dogs or fewer. The gray or black color class was almost exclusively present in the low PC1 subpopulation and GSDs from kennels with high working proportions were more common in the low PC1 subpopulation (Supplementary Fig. 4). It is generally known that working type GSD more often are of gray/black colors compared to the show type. Based on these two levels of support, we concluded that the split

across PC1 is most likely explained by a breed type division into working and show type GSD.

**Sample collection and genotyping**. We retrieved genotype data from the Illumina CanineHD 170 K BeadChip genotyping array (Illumina, San Diego, CA) generated from blood samples from dogs collected from privately owned dogs in collaboration with several veterinary clinics throughout Sweden (LR, GR, GSD, and WHWT), US (WHWT), and Switzerland (LR and GR). The Swiss cohort included samples from dogs collected in Switzerland, Netherlands, Finland, Germany, and France. Dogs were recruited to the project as their owners visited the veterinary clinic to seek health care for AD (cases) or unrelated problems (controls), or were recruited as healthy controls followed by a visit to the veterinary clinic to leave blood samples. This applied to all countries except the UK. Saliva samples from the UK (LR and GR) were collected by owners of the dogs and posted to the research team as part of the questionnaire study[64], and genotyped by Neogen using the Illumina CanineHD 230 K BeadChip (Illumina, San Diego, CA). Samples for each cohort were collected strictly according to regulations defined by each country. Ethical approval for the UK project was provided by the University of Nottingham School of Veterinary Medicine and Science Committee for Animal Research and Ethics. Protocols for US dogs were approved by the North Carolina State University Institutional Animal Care and Use Committee. Collection of the samples and clinical data from Swiss dogs was approved by the Cantonal Committee for Animal Experiments (Canton of Bern; permits 22/07 and 23/10) and from the Swedish dogs by ethical permit C12/15.

The CanineHD 230 K BeadChip is an extension of the CanineHD 170 K BeadChip and we started with a merged genotype dataset consisting of 167,211 SNPs and 1152 dogs from four dog breeds. Genomic coordinates refer to the canFam3.1 genome assembly unless otherwise specified. We used plink (v. 1.90b4.9)[100] and R (v. 4.9.2)[101] with the following R-packages GENESIS (v. 2.24.0)[102], GWASTools (v. 1.40.0)[103], and SNPRelate (v. 1.28.0)[104] to analyze the genotyped datasets separated by breed. QC was performed per dog breed (plink --geno 0.05 --mind 0.05 --maf 0.05). Genetic relationship was estimated using the KING method of moment for the identity-by-descent analysis[105] in SNPRelate. Individuals with a kinship coefficient above 0.177 (~2nd degree relatedness) were removed to generate a dataset with highly related dogs excluded. PCs were estimated using pcair (part of the GENESIS R-package) with the following settings for snpgdsLDpruning: method = r, ld.threshold = 0.7, slide.max.bp = 250000, maf = 0.05, missing.rate = 0.05, and for pcairPartition and pcair: kin.thresh = 0.125, div.thresh = −0.125. In pcair, the PCs were estimated in a subset of individuals unrelated at the kinship coefficient threshold of 0.125, after that step PCs were projected on the more related individuals (i.e., 0.125–0.177). This was to avoid bias due to relatedness in the PC estimation. The individual filtering and PC estimations was performed on the original genotyped datasets, and the resulting sample set and PCs were used for downstream analyses using imputed markers. R-package qqman (v. 0.1.8)[106] and Adobe Illustrator (v. 26.0.2) were used for plots and final editing of Figures.

**Imputation**. We imputed the genotyped dataset using a reference panel of purebred dogs (435 individuals) extracted from a publicly available dataset including wolf and other canids[90]. Imputed datasets have better genome coverage, which in the association study gives an improved sensitivity and precision when detecting candidate regions, and increases the likelihood that important regions are not missed. One problem with imputation may be that unique haplotypes are not covered by the reference panel; however, the risk is small given the extensive reference panel with our studied breeds included and all genotyped markers are still included. Quality parameters used in plink prior to imputation were --maf 0.0001, --geno 0.05, and --mind 0.05 and the dataset before imputation consisted of 1152 dogs (347 LR, 294 GR, 231 GSD, and 280 WHWT) and 148,889 SNPs. Imputation was performed as follows: (i) the data was split into each chromosome (except for chromosome 1, which was split into two parts) in plink while filtering on --maf 0.001 and --geno 0.2. (ii) we used SHAPEIT2 (r904)[107] (--check) to check if SNP genotype data existed in the reference panel and SNPs not found were excluded in the next step (N = 9529). (iii) the genotype data was pre-phased with the reference panel using SHAPEIT2 (with settings effective-size 500, details on the Markov chain Monte Carlo iterations were --burn 10 --prune 10 --main 50, and threads -T 5). (iiii) we used IMPUTE2 (v. 2.3.2)[108] to impute the genotype data (--Ne 500). We used SHAPEIT2 to check SNPs on chromosome X (after using plink --split-x 6,600,000 123,798,852[109], --maf 0.001 and --geno 0.05 for chromosome X specifically) and identified problematic dogs with high rates of heterozygosity on chromosome X ( > 1%). These dogs were removed when merging all chromosomes (chromosome X not included) after imputation ending up with 336 LR, 287 GR, 229 GSD, and 275 WHWT.

**Dataset quality and details**. After imputation, the dataset was split into each dog breed and analyzed breed-wise (QC: plink --geno 0.02 --mind 0.05 --maf 0.05). We used plink to LD-prune the imputed datasets (--indep-pairwise 25 5 0.999) followed by adding all genotyped SNPs that were excluded in the pruning step. A second step of LD-pruning was performed in GR and LR using a stricter threshold

(--indep-pairwise 50 5 0.99) since the total number of variants from the first step of LD-pruning exceeded fastPHASE (v. 1.4.8)[110] capacity (Supplementary Table 2).

**Imputation validation**. IMPUTE2 automatically produces a concordance table of the internal cross-validation. The program masks genotypes of one variant at a time and imputes the masked genotypes, and compares imputed genotypes with the original genotypes. The provided concordance rate for each chromosome after imputation ranged from 96.2% to 98.8%. As an additional validation of the imputation quality, we randomly masked 5,000 SNPs in the genotyped dataset (3.36% of the total SNP set). Imputation was performed as described above, and out of the masked SNPs, 3,509 SNPs were imputed (70.2%) with a concordance rate of 99.5% across all four breeds, including chromosome X (plink --merge-mode 7). Excluding chromosome X left 3,408 SNPs for validation resulting in a concordance rate of 99.5% in all breeds together. Extraction of data per breed and filtering on --maf 0.05 resulted in concordance rate in LR: 99.4% (out of 2,453 imputed SNPs), GR: 99.6% (2,184 SNPs), GSD: 99.6% (2,036 SNPs), and WHWT: 99.7% (2,025 SNPs). If no maf filter was applied, the concordance rate ranged from 99.4% to 99.6% across the breeds.

**Bayesian mixture model**. We used the BMM BayesR (v. 1, update 01/04/2021)[20,21] to perform a GWAS in each breed separately. The BayesR algorithm estimates the probability that a variant's effect size belongs to either of the following four normal distributions: N(0, 0σ²ₒ) i.e. zero-effect, N(0, 0.0001σ²ₒ), N(0, 0.001σ²ₒ), or N(0, 0.01σ²ₒ). The proportions of variants belonging to each distribution are updated in each iteration. The model was run with 300,000 iterations and 100,000 burn-ins to achieve optimal convergence and was also repeated five times. The absolute value of the average effect size per variant was reported as the final result. Fixed effects were the first two (LR and WHWT) or three (GR and GSD) PCs (defined by fitNullModel in GENESIS to have significant (p < 0.05) effect on the trait). For GSD, the -log₁₀ of IgA levels and -log₁₀ of age in years at sampling were included as fixed effects in line with the described relationship between AD and low serum IgA levels in GSD[12]. To determine a rational cutoff for defining effect variants throughout all four populations of breeds, we chose the value of 0.0001 from the lowest effect size distribution and applied this for all breeds to generate comparable results. Therefore, we regarded variants with mean absolute effect size larger or equal to $1.00 \times 10^{-3}$ as effect variants. Effect variants separated by >1 Mb were considered to belong to separate associated loci and the effect variant with the highest absolute effect size for each locus was extracted to represent the associated locus. A risk index was calculated by quantifying the number of risk genotypes from each associated locus (0 = no risk allele, 0.5 = one risk allele, 1 = two risk alleles).

**Selection signature analysis**. We performed whole genome analysis for signatures of selection by comparing AD cases with controls within each of the four dog breeds according to the EHH concept[111]. Haplotypes for case and control populations were identified with fastPHASE (v. 1.4.8)[110], using default settings except for the number of random starts set to 10 (-T10). Next, we applied rehh (v. 3.2.1)[112] to calculate the XP-EHH by comparing the iHH between the case and control population at each variant position[22]. Signatures of selection were identified when one population had overrepresented and extended haplotypes compared to the other population. When the case population had extended haplotypes compared to the control population the XP-EHH value was positive, and vice versa. The calc_candidate_regions function in rehh was used to define the major regions under selection with the cutoff for extreme markers at -log10(p) XP-EHH ≥4, same as the default (all settings: threshold = 4, pval = TRUE, window_size = 1E6, overlap = 1E5, min_n_extr_mrk = 2). Canine TADs[28] from liver were also used to characterize XP-EHH regions that are presented in more detail in the main figures. A TAD is described as a self-interacting genomic region where DNA sequences within a TAD physically interact with each other more often than with sequences outside the TAD. Thus, studying TADs may give indications of which genes are more likely to be regulated by different variants. The TADs can differ between tissues but the canine liver TADs can give us an estimation of what genes may be regulated in the nearby region to the variants defined in this study, even if the more appropriate tissue would have been skin. We used plink --assoc (1df chi-square allelic test) or --logistic (logistic regression with covariates) to evaluate if variants around a selection signal (i.e., estimated by the EHH plot) were associated with canine AD. Associated variants from plink assoc (chi-square allelic test) and/or logistic regression models were regarded as potentially associated with AD. Haploview (v. 4.2)[113] was used to evaluate potential LD blocks in the region on chromosome 3.

**Characterization of canine AD regions**. Genes in candidate regions shown in Figs. 3–5 were extracted from canFam3.1 public track hub Broad Improved Canine Annotation v1[114]. For better visualization, the longest transcript per gene was kept and transcripts named ENSCAFG or CFRNASEQ_PROT (lacking official gene symbol nomenclature) were removed. For main tables, gene transcript information was extracted from the canFam3.1 genome assembly but we also used the UU_Cfam_GSD_1.0/canFam4[31] (canFam4) annotation to provide additional information and update transcript information.

To investigate more distant potential candidate genes in AD-associated loci, we extracted protein coding genes located within 1 Mb (the approximate size of a TAD[115]) from effect variants of each associated locus and denoted these BayesR regions. For selection, genes within XP-EHH regions were extracted. STRING (v. 11.5)[116] was used to evaluate gene set enrichment and potential interactions across loci. Both *Homo sapiens* and *Canis lupus familiaris* were used as background models for evaluating the genes in BayesR regions, XP-EHH regions, and in both sets combined. Enrichment terms with one region represented were regarded as non-relevant for evaluation of enrichment across loci. Specific terms with many regions represented and with relevance to canine AD were highlighted. In addition, all genes located in BayesR regions (+/−1Mb from effect variants) and in XP-EHH regions (Supplementary Data 10-11), were compared to associated genes from human GWAS of dermatitis, atopic eczema, eczema and psoriasis from the GWAS catalog[117] in order to detect gene overlaps between human skin disorders and canine AD.

The phyloP score is the log p-value under a null hypothesis of neutral evolution, and a positive score indicates evolutionary conservation where positions in the genome remain the same across many species because they are functional. In contrast, negative phyloP scores indicate accelerated evolution, potentially corresponding to positive selection. Genomic positions with phyloP scores >2.56 were considered evolutionary constrained at FDR < 5% (240 species[29]). We considered the phyloP scores for the variants defined as extreme markers (-$\log_{10}$(p) ≥4.0) in candidate regions of selection, for all effect variants, and variants in LD with effect variants in the chromosome 17 locus. The effect variants were intersected with BarkBase ATAC-seq data[23], and, for variants lifted to hg38, ENCODE cCREs[24–26] (from UCSC Genome Browser) and GeneHancer[27] elements. LiftOver[118] between genomes (canFam3.1, canFam4, and human hg38) was used to evaluate and compare functional and non-functional positions across assemblies.

**Oxford Nanopore Technologies whole genome sequencing**. Two LR AD cases (ID1 and ID2) and two LR controls (ID3 and ID4), heterozygous risk and homozygous non-risk for chromosome 17 effect variants respectively, were chosen for ONT long-read sequencing. DNA was extracted from EDTA blood samples from the four dogs using the NucleoSpin Blood kit (ref 740951, Macherey-Nagel) following the standard protocol, with the exception that DNA was eluted in 50 μl $H_2O$ instead of Buffer BE, followed by incubation at RT for 3 min and centrifugation for 1 min at 11,000xg. The elution step was repeated once. DNA concentration was checked by Qubit (Invitrogen, Thermo Fisher Scientific), and DNA size and integrity was assayed using a Genomic DNA Screen Tape (Agilent). DNA was fragmented with g-TUBEs (Covaris) resulting in an average fragment size of 6 kb. The fragmented DNA was prepared for sequencing using the MinION SQK-LSK109 kit (ONT) following the protocol except for two minor differences. For ID1 and ID3, the AMX-F adapter mix was used, whereas for ID2 and ID4 the AMX adapter mix was used. The DNA library for sequencing was loaded and run on four separate R9.4.1 SpotON flow cells (ONT). The.fast5 files were base-called with the Super accurate model in Guppy (v. 6.0.1) (ONT). FASTQ files were mapped to canFam4 with minimap2 -x map-ont[119,120]. Variants were called using clair3[121,122] and SVs were called with Sniffles (v.2.0.3)[123] using phased BAMs and the --phase command, --tandem-repeats to define repeat regions in the reference genome, and --reference canFam4. SVs were analyzed in windows of 11 bp with start position (indicated by Sniffles) in the middle of the window (because the exact start position could vary with a few bp between samples) and end position (indicated by Sniffles) was reported as the exact position. The windows in all four samples were intersected with bedtools intersect[124].

Variants on chromosome 17 following the same pattern as the effect variants on chromosome 17 (heterozygous cases and homozygous non-risk controls) were extracted and evaluated in the non-LD pruned imputed dataset. All variants in LD ($r^2 > 0.8$) with any of the effect variants on chromosome 17 were extracted for further evaluation. Variants not included in imputed data were also extracted and defined as novel. Two effect variants were excluded based on evaluation of sequence data (Supplementary Note 1 and Supplementary Figs. 8 and 9). The four sequenced dogs were also evaluated individually using read-based haplotype phasing focusing around the chromosome 17 effect variants. Phasing was performed using WhatsHap (v. 1.2.1)[125] within the clair3 pipeline and haplotypes were reconstructed with the bcftools[126] (--consensus command). In IGV[127], the phased BAM files were tagged by and sorted on haplotype. Genotype and haplotype assignment for the effect variants on chromosome 17 in LR were identified in the phased BAMs for all four individuals. Potential functionalities of variants were evaluated by SnpEff (v 4.3.t)[30] (canFam4 and a custom built database with the NCBI annotation of the reference), phyloP[29], BarkBase ATAC-seq data[23], CpG annotation in camFam4, and, for variants lifted to hg38, ENCODE cCREs (from UCSC Genome Browser) and GeneHancer[27] elements. LiftOver[118] between genomes (canFam3.1, canFam4, and human hg38) was used to evaluate and compare functional and non-functional positions across assemblies.

To extract regions of low heterozygosity (i.e., regions where phasing failed), the phased reads for the two controls and the two cases in region canFam4 chr17: 55,000,000–63,000,000 were filtered on assigned haplotype tag HP1 and HP2 using bamtools[128] filter. Reads without HP tag common for both controls or both cases were extracted using bedtools[124] intersect and merged in regions if overlap within 1 kb start/end with bedtools merge. Regions unique to the controls were extracted with bedtools intersect.

**Statistics and reproducibility**. Comparisons between case and control populations for risk index were performed using Welch Two Sample T test (two-tailed) and boxplot (R package stats v. 4.1.2[101] and graphics v. 4.1.2[129]). Differences in allele frequencies between cases and controls within each breed type of LR and GSD was calculated using Pearson's Chi-squared test with Yates' continuity correction (R package stats v. 4.1.2). We calculated phenotypic variance explained by AD-associated loci using a linear model and ANOVA (R package stats v. 4.1.2), and included PC1-2 in the analysis of LR and WHWT, PC1-3 in GR, and PC1-3, -$\log_{10}$(IgA), and -$\log_{10}$(Age) in GSD (same as in BayesR). The total sample sizes for BayesR and XP-EHH analyses were 321 LR, 256 GR, 219 GSD, and 235 WHWT, and the same sample sizes were used in the additional statistical tests except for the chi-squared test of breed types of LR and GSD. The total sample sizes for breed types were; 245 common type LR, 76 gundog type LR, 110 show type GSD, and 107 working type GSD.

**Reporting summary**. Further information on experimental design is available in the Nature Portfolio Reporting Summary linked to this Article.

## Data availability

Genotype datasets (plink files: bed, bim, fam, pheno) per breed after quality controls and relatedness filtering as well as the combined dataset (bed, bim, fam) used for imputation have been uploaded in SciLifeLab https://doi.org/10.17044/scilifelab.21287139.v1. The sequencing data of four Swedish LR were deposited to ENA with accession number for project: PRJEB55514, study: ERP140417, and samples: ERS10220104- ERS10220107. Source data for Figs. 2, 4i, and 6 are found in Supplementary data 2, 8, and 9, respectively.

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

## Acknowledgements

Thank you to all the dogs and their owners who took part in this research. The UK samples were collected and curated as part of the Itchy Dog Project, for which funding was provided by Dogs Trust as part of their Canine Welfare Grants scheme. The UK project team included Dr P Craigon, Professor G England, and Dr S Shaw who assisted with sample collection, project management, and provided clinical expertize. N.J.O. and T.O. thank the American Kennel Club Canine Health Foundation and the Westie Foundation of America for their support in the sample collection and first GWAS analysis in WHWT in the US. We would like to thank Mats Pettersson for providing statistical expertize. We also thank Marcin Kierczak from National Bioinformatics Infrastructure Sweden at SciLifeLab for valuable discussions on statistical methods and for comments on the manuscript. Computations and data handling were enabled by resources in projects, SNIC 2017/7-384, SNIC 2017/7-385, and SNIC 2021/5-579, provided by the Swedish National Infrastructure for Computing (SNIC) at UPPMAX, partially funded by the Swedish Research Council through grant agreement no. 2018-05973. This project was funded by the European Research Council (LUPA project, GA-201370) to K.T., E.S., C.W., O.W., E.P., Å.K., J.R.S.M., P.R., T.L., and K.L.T., the Swiss

National Science Foundation (310030_200354) and the Albert-Heim Foundation to T.L., AKC Canine Health Foundation with support from the West Highland White Terrier Club of America to N.O., and the Swedish Research Council to K.T., E.S., C.W., O.W, E.P, Å.K., J.R.S.M., and K.L.T.

## Author contributions

K.T., K.B., Å.H., G.A., and K.L.T. participated in conception or design of the work. K.B., O.W., E.P., Å.K., N.D.H., S.C.B., N.O., T.O., P.R., and T.L. in the acquisition of samples and data. K.T., E.S., C.W., K.B., O.W., G.B., J.R.S.M., and K.L.T worked with analysis and/or interpretation of data. K.T. and E.S. drafted the work and C.W., T.O., T.L., K.L.T., and G.A. substantively revised it.

## Funding

## Competing interests

The authors declare no competing interests.
