## [Peer Review File · Communications Biology]

Reviewers' comments:

Reviewer #1 (Remarks to the Author):

The authors did excellent work to detect canine atopic dermatitis (AD) associated genetic variants in four breeds using novel GWAS (BayesR) and haplotype-based selection signature (XP-EHH) methods, which revealed a bunch of valuable genetic risk loci. Especially several identified genes could be confirmed through across methods, which highly convinced me the statistical power in this study. In short, no matter from scientific writing, experiment design, statistical analysis, results interpretation. I only have several minor comments/questions to discuss with the authors:

Minor comments/questions:

On the abstract, the abbreviation for canine atopic dermatitis is AD, while on the introduction is CAD. Might be better to be consistent.

For the last paragraph on the introduction part, the authors indicated the purpose and brief outcome of the current study. It would be better also shortly to introduce something extra for the dataset, breeds etc.

For the sample collection and genotyping part, if it was possible to add one sentence to describe the number of overlap SNPs between 170K and 230K chips? I don't find enough information from the Illumina website for the 230K chip and the extra information might be helpful to future readers.

For the imputation and imputation validation, I am not sure if it was also possible to just use only the overlap between 170K and 230K SNP chips? Or if the imputation could benefit much to later data analysis (GWAS and XP-EHH) comparing only using the overlap between 170K and 230K SNP chips? Several sentence to further explain why this study decide to imputation might be good enough.

For the selection signature, not clear how do the authors decide the final size of 'windows'? I guess, something like calculate the values for each locus first and afterwards manually combine loci generally with high values into the regions?

Reviewer #2 (Remarks to the Author):

Tengvall et al. utilized more than 1000 dogs including both cases and controls from 4 dog breeds (LR, GR, GSD and WHWT) to study the complex canine atopic dermatitis (AD) disorder. Bayesian genome-wide association and selection signature detection (XP-EHH) methods were applied. 15 significantly associated loci were identified by BayesR, 8 candidate genome regions were detected above the XP-EHH threshold. One interesting region on chromosome 17 which is also the risk locus in human AD was identified to be associated with AD. Further bioinformatic analysis of the identified candidate genes revealed some of them were involved in immunity or epidermis, which supports the hypothesis by considering the gene functions. Since the canine AD is a complicated disease, it is not easy to identify causative variants. This paper provides important potential clues for figuring out genetic risk factors of dog AD disease.

Comment#1

In line 62-63, several different versions of reference genomes were used, please transfer all positions in CanFam4 by Gemome Liftover to make them consistent.

Comment#2

In line 116, for Bayesian GWAS, the exact number of cases and cases of each breed was not given. Please also give the sample information details for XPEHH analysis.

Comment#3

In line 134, you could check if there are epigenomic peaks in Barkbase for the candidate associated chr17 region.

Comment#4

If there are some common variants that were detected by both BayesR and XPEHH?

Comments#5

In line 191, you could check the variant effect for the LRP1B missense variant by SIFT, PolyPhen-2 or PROVEAN.

Comment#5

In line 997, one column could be added to give information about the distance between the associated variant and the gene. The variant frequency in cases and controls should also be indicated in the table.

Comment#6

The identified candidate genes could be searched in human GWAS database (<https://www.ebi.ac.uk/gwas/>) to see if they have some associations with human AD.

Comment#7

Body weight genes like TBC1B1 and HMGA2 were detected in this study, but dog body size genes were strongly selected by human. HMGA2 is strongly selected in small sized dogs. How could you tell if the detected signals are associated with AD rather than body size? Or is there a possibility to consider body size as a co-variate?

Reviewer #3 (Remarks to the Author):

In "Revealing risk factors for Canine Atopic Dermatitis using Bayesian model and selection signature analyses", the authors use an innovative approach to gain more insight into AD in dogs. Instead of using a traditional LMM GWAS, they suggest a Bayesian approach to account for the polygenic nature of the disease and to identify risk factor alleles rather than single high effect loci. I really like the complementary GWAS/ XP-EHH method as an attempt to investigate the hypothesis of potential hitchhiking. Following the Bayesian test, which identified a major risk locus on Chr17 for LR, the authors conduct some sequencing to identify further variants associated with AD and give a comprehensive picture for this risk locus.

In general, it's an interesting study with sound methods and I'm impressed by the complexity/ the questions the authors have addressed to investigate the genetic basis of AD in dogs. There are two major points I'd like the authors to address:

First, readability is difficult at times (due to a lack of information/ explanation in Material & methods specifically and throughout the manuscript).

Second, grouping dogs into working/ show subpopulations based on PCA results is questionable and introduces bias in my opinion.

Regarding the first point, I'm commenting on specific examples in the following, but I would encourage the authors to revise their manuscript in general with readability in mind.

- L492: "Swedish LR (...)" in title, but first sentence says "Dogs from LR breed (...)", which implies it's all LRs. This is very confusing throughout the whole manuscript, it appears you only grouped the Swedish LRs in show/ working, but when you look at the numbers (e.g. Figure 6) it is LRs from all countries (LRs add up to 322 here in the different groups, but according to L499-500 you could only group 33 and 51 LRs into the respective groups). Also Figure 6, there is no group for missing data (at least 19 LRs from Sweden with no information)? I later realised that you used the PCA to group all

dogs into the subpopulations, but this is not properly explained anywhere. The same applies for GSDs.

- L520: You describe sample-wise QC here (e.g. relatedness), but what about QC on genotype data? I understand that you filter for maf and genotype rate later, but there are also other parameters that are usually filtered for (e.g. Gene train score and signal intensity (ABRmean) in GenomeStudio).

- L524-528: It is good to give the exact specifiers used, but it is also helpful to explain what they are doing for people unfamiliar with the used tools. It is for example not very clear to me, if you have done the PCA on the genetic relationship matrices? Also, in which step do you remove samples (I assume "kin.thresh" filters for samples with relatedness < 0.125)?

- L531-551: I'd suggest to explain the steps in more detail (i.e. describe what the different specifiers mean, as mentioned before). It is for example not obvious to everyone that you prune the data in iii) with "--prune" and L549 poses the question where the first step of pruning was.

- Supp Table 1: In the text, Supp Table 2 is mentioned before Supp Table 1. As described later, some numbers are confusing. Would it maybe make more sense to restructure the data description?

- L668: Why did you fit different factors for different breeds?

Results: I understand that the journal style requires the Results section before Material & methods. I would suggest to add some more information in the results section to help readers understanding what you have done.

- L118: Please explain how loci/ regions were determined (also in methods).

- L120: Why this threshold of effect size for effect loci?

- L129: Principal component 1, explain here what PC1 is (from PCA on genotypes), what does PC1 characterise?

Generally, Tables and Figures need more comprehensive explanations and legends

- Supp Table 5: "Residual" missing

- Why Supp Table 5 and 6 separate?

- Supp Table 7: Explain headers

- Table 2: Are statistics given as average for region?

- Supp Fig 2: XP-EHH between which populations? Clarify that it's cases vs. controls

- Supp Fig3: Use show/ working instead of "field" for consistency

- Fig4/L1045: This is only Swedish LR? Was the PCA only performed for Swedish LR? XP-EHH only for Swedish dogs?

- Supp Table 4 and 5: Why no PC3 fitted?

- Supp Table 6: Why two sub-tables for WHWT? Why no PC3 fitted for WHWT? Explain whole table better (e.g. that loci fitted are risk loci)

- L481 and Supp Table2: 67 and 57 in text vs. 62 and 48 cases in UK LR and GR in Table (which number is correct?). The same non-matching numbers for controls. I assume it is number of dogs with phenotype and genotype in Table, whereas it is only dogs with phenotypes in text. Maybe make this clear in the Table and also mention this in text when you make the reference to Supp Table 2 (491)?

- Table 3: "XP-EHH" instead of "Selection". How was function "immunity" or "epidermis" assigned?

- Supp Figure 1: Needs more explanation in the figure legend (it is PC scores, why relationship matrices?)

- Figure 4a/ Supp Figure 3: it's both PCA on Swedish LR, why are there differences in the plot? Again, I now understand that Figure 4 is all dogs and Supp Figure 3 is only Swedish dogs, but as a reader you think you only have dog type information on Swedish LRs (because you don't describe your PCA approach to group dogs), which leads to a lot of confusion throughout the manuscript.

Regarding the second point, I had initial problems to understand how you only have dog type information for some Swedish LRs and some GSDs, but perform comparative analyses grouping all LRs and GSDs into show/ working. As far as I am aware, you only explain in the legend of Figure 5 that you use the PCA results and where the Swedish LRs and GSDs with known dog types cluster to decide that scores for PC1 discriminate between dog types. Clusters can be attributable to other factors (e.g. country in case of LR, coat colour etc.) and in my eyes you don't have a strong case for a confounding between dog type and AD based on your assumptions/ methods. I would suggest to a) only include LRs and GSDs for which the dog type is known to compare risk alleles or cases/ controls in

subpopulations or b) group dogs into high vs. low PC1 scores and discuss later what PC1 might be associated with.

- Figure 5/ L216: If I understand this correctly, you group all LR into working/ show based on the PCA scores for Swedish LR? I am not convinced that this is a suitable approach, as stated above, PC1 could also capture variation explained by "country". Using your assumption, you'd conclude that all NE and Suisse dogs are show dogs, is this likely?
- Figure 6: You report information for 217 GSDs (why does it say 219 dogs in Supp Table 2?), and I assume you have again used PCA to split dogs into show/ work? In GSDs, population structure can also be confounded with coat colour (working dogs darker phenotype).
- L1046: "thus we can conclude that the division formed in the relationship matrix can be attributed to working type LR in the low PC1 cluster and show type LR in the high PC1 cluster": But you only have this information for Swedish dogs, i.e. a high number of "points" must be "NA" (it is not very clear from the points size). When you look at Supp Fig 1, you could conclude that high scores in PC1 are attributable to dogs from NE and Switzerland, hence indicating a confounding with country rather than with working/ show dog?
- L359: Interesting point. However, as I mention earlier, the two subpopulations indicated by PC1 might be explained by attributes other than dog type (especially since you don't have information for all dogs under analysis). Label groups high vs. low scores for PC1 instead of "working vs. show" and try to disentangle what factor might be confounded with PC1 (e.g. dog type, coat colour, country in LR). You suggest here that body weight is what differentiates the two subpopulations, if that's the case one conclusion could also be that there is a difference between countries, e.g. that NE and Switzerland select for heavier dogs.
- L400: Again, interesting point. I think it would be more significant if comparison happened only between GSDs for which working/ show status was known (excluding all others instead of extrapolating them based on PCA results).

Other points:

Apologies that the comments are not entirely in order, but that's the order I read through the manuscript.

L62-63: I'd like to positively highlight the author's efforts in being very precise with the assembly version to enable comparisons between studies.

Please consider rephrasing L68-83: You specify limitations (not assumptions) of GWAS, which can be solved with appropriate data (e.g. small effect sizes can be detected when sample size is large enough, relatedness can be accounted for by fitting relationship matrix/ remove highly related samples), WGS is not an approach (you can do a GWAS with WGS, also, by only sequencing a couple of samples you won't be able to identify rare variants):

- L68-71: Maybe reconsider your description here? I think what you're trying to say is that in LMM, you test for associations between a single marker and the trait and pick up only markers with high effect size, whereas with a Bayesian approach, you test the joint effect of a set of markers (predict the risk for AD based on these loci).
- L76: You can do a GWAS with WGS data? Or is what you mean that instead of SNP array genotyping WGS is able to identify rare variants?
- L77: what do you mean with "evolutionary constraint variants"?
- L78-83: Use genomic relationship matrix? As long as you have genotype data you can calculate relationship between samples and include it in your model. (see here <https://www.nature.com/articles/ng.3190> "Linear mixed models are emerging as the method of choice for association testing in genome-wide association studies (GWAS) because they account for both population stratification and cryptic relatedness and achieve increased statistical power by jointly modeling all genotyped markers")

L99: Remove ",,"

L462: How were the dogs chosen/ invited to participate? You explain in the Reporting Summary it's dogs (or better their owners) seeking health care, which I would include in Material and methods

L489: What about phenotyping for remaining countries (e.g. Germany, Netherlands, France)?

L499: According to Supp Table 2, it's 102 LRs with Swedish origin.
L525: Give reference for p_{cair}
L534: Mention that you impute to WGS
L539: Is there something missing here "breed-dogs"?
Supp Table S1 and S2: Total number of LR differs (321 vs 322).
L568: Change to "belongs"
L578: Why was this threshold chosen?
L579: Clarify what you consider as region.
L591: How does the function define regions of selection?
L592: What do you consider as "top selection loci"?
L591: Explain TADs and what you did here.
L595: What do you mean by "variants forming clusters of association"?
L607: Why did you use this distance?
L614: canine AD instead of CAD (because you use canine AD throughout the rest of the manuscript)?
L616: Explain phyloP scores (e.g. score that describe evolutionary conservation)
Is there a difference between working/ show comparison and field/ no-field comparison? If not, stick to one terminology to avoid confusion.
L115: Maybe better "(...) identifies fifteen risk loci for canine HD"?
L116: There are also dogs from other countries included according to Supp Table 2?
Supplemental tables: use "." as decimal separator instead of ","
L134: Why was the loci on Chr 17 (with 9 effect variants) chosen and not Chr 36 (with 10 effect variants)?
L145: Change to "overlapped with"
L156: Mention that this is the human reference genome
L185: how do you define "candidate region of selection"?
L185: Which three breeds?
L195: It's "low scores for PC1"
Figure 4a: Differentiation between dog types is really hard to see in figure, consider using different symbols instead of point sizes?
L200: "allele C was more frequent in show subpopulation" where is this shown in Figure 4?
L260: Please consider using "Genes under putative selection" instead of "selection genes" throughout the manuscript
L285: Rephrase this sentence, e.g. name the disease you found associations for.
L286: What was the reason for choosing these cut-offs (not explained in Material and methods)? Are there comparable values reported in other studies?
L293: According to this argument, would you expect "better" results for array data instead of WGS data? What was the motivation of using WGS data, considering that this goes along with certain bias (due to imputation) and you had to prune the data afterwards anyway due to computational restrains?
L296: Clarify that it is a risk loci for LRs
L308 and following: Give gene names in italic
L358: Do you have investigated the coat colour in your dogs (e.g. for clustering in PCA, for association with risk loci/ risk index)?
L365: Clarify that it is a risk loci for GSDs
L400-402: Please clarify what you mean here.
L520: started with merged data set of all genotyped dogs, why different genotype number for breeds in Supp Table1? I assume you applied the QC within breeds?

Reviewers' comments:

Reviewer #1 (Remarks to the Author):

The authors did excellent work to detect canine atopic dermatitis (AD) associated genetic variants in four breeds using novel GWAS (BayesR) and haplotype-based selection signature (XP-EHH) methods, which revealed a bunch of valuable genetic risk loci. Especially several identified genes could be confirmed through across methods, which highly convinced me the statistical power in this study. In short, no matter from scientific writing, experiment design, statistical analysis, results interpretation. I only have several minor comments/questions to discuss with the authors:

Minor comments/questions:

On the abstract, the abbreviation for canine atopic dermatitis is AD, while on the introduction is CAD. Might be better to be consistent.

#R: Fully agree, this is corrected now.

For the last paragraph on the introduction part, the authors indicated the purpose and brief outcome of the current study. It would be better also shortly to introduce something extra for the dataset, breeds etc.

#R: Yes. We now added in the following:

L:111 "We performed genetic mapping in four dog breeds predisposed to AD, using datasets consisting of samples from ~200-400 dogs per population in a joint international collection."

For the sample collection and genotyping part, if it was possible to add one sentence to describe the number of overlap SNPs between 170K and 230K chips? I don't find enough information from the Illumina website for the 230K chip and the extra information might be helpful to future readers. For the imputation and imputation validation, I am not sure if it was also possible to just use only the overlap between 170K and 230K SNP chips? Or if the imputation could benefit much to later data analysis (GWAS and XP-EHH) comparing only using the overlap between 170K and 230K SNP chips? Several sentence to further explain why this study decide to imputation might be good enough.

#R: 230K chip is an extension of 170K, all markers in 170k should be included in 230K. Both chips are from Illumina, but only the 170k is introduced on their website. We used to get the 230K genotyping result from Neogen, but it seems this service is no longer available. Information from an old webpage about this 230k genotyping says:

"The Illumina CanineHD BeadChip offers more than 230,000 evenly spaced SNP's in the CanFam3.0 assembly. The BeadChip was originally developed in collaboration with the LUPA Consortium, which includes 22 European universities and other partners such as the Broad Institute."

We now added the following sentence for clarification in Materials and Methods:

L:627 "The CanineHD 230K BeadChip is an extension of the CanineHD 170K BeadChip and we started with a merged genotype dataset consisting of 167,211 SNPs and 1,152 dogs from four breeds."

L:646 "We imputed the genotyped dataset using a reference panel of purebred dogs (435 individuals) extracted from a publicly available dataset including wolf and other canids⁹¹. Imputed datasets have better genome coverage, which in the association study gives an improved sensitivity and precision when detecting candidate regions, and increases the likelihood that important regions are not missed. One problem with imputation may be that unique haplotypes are not covered by the reference panel used for imputation; however, the risk is small given the extensive reference panel with our studied breeds included and all genotyped markers are still included."

And added the following to Discussion:

L: 322 "On the other hand, the genomic positions indicated by effect variants are more precise and the risk of missing important regions associated with the trait is decreased in a denser marker set."

For the selection signature, not clear how do the authors decide the final size of 'windows'? I guess, something like calculate the values for each locus first and afterwards manually combine loci generally with high values into the regions?

#R: We think you refer to this part "rehh:calc_candidate_regions (using default settings)" for selection analysis. The settings in rehh we used were: threshold = 4, pval = TRUE, window_size = 1E6, overlap = 1E5, min_n_extr_mrk = 2.

For clarity we changed the sentence in Materials & Methods:

L:719 “The function `rehh:calc_candidate_regions` was used to define the major regions of selection with the cutoff for extreme markers at $-\log_{10}(p) \geq 4$, same as the default (all settings were: `threshold = 4`, `pval = TRUE`, `window_size = 1E6`, `overlap = 1E5`, `min_n_extr_mrk = 2`).”

We now also clarified in Results:

L:192 “In LR, GSD, and WHWT, a total of eight candidate regions of selection (XP-EHH regions) were identified. Regions were defined using a 1Mb window scan with 0.1Mb overlap and at least two variants with $-\log_{10}(p)$ XP-EHH above 4 (Fig. 1, Supplementary Fig. 2, Table 2).”

Reviewer #2 (Remarks to the Author):

Tengvall et al. utilized more than 1000 dogs including both cases and controls from 4 dog breeds (LR, GR, GSD and WHWT) to study the complex canine atopic dermatitis (AD) disorder. Bayesian genome-wide association and selection signature detection (XP-EHH) methods were applied. 15 significantly associated loci were identified by BayesR, 8 candidate genome regions were detected above the XP-EHH threshold. One interesting region on chromosome 17 which is also the risk locus in human AD was identified to be associated with AD. Further bioinformatic analysis of the identified candidate genes revealed some of them were involved in immunity or epidermis, which supports the hypothesis by considering the gene functions. Since the canine AD is a complicated disease, it is not easy to identify causative variants. This paper provides important potential clues for figuring out genetic risk factors of dog AD disease.

Comment#1

In line 62-63, several different versions of reference genomes were used, please transfer all positions in CanFam4 by Genome Liftover to make them consistent.

#R: We chose to report the genome assembly version used for the respective study (since the studies were published at different time points they had used different canine genome versions) to avoid confusion if readers want to look up the references. Reviewer #3 proposed that the reporting of genome assembly version was an appropriate way to enable comparison between studies: “L62-63: *I'd like to positively highlight the author's efforts in being very precise with the assembly version to enable comparisons between studies.*” Since we see no overlap with any of these studies, we do not believe it is necessary to discuss them further.

Comment#2

In line 116, for Bayesian GWAS, the exact number of cases and cases of each breed was not given. Please also give the sample information details for XPEHH analysis.

#R: We agree that information in the Results should be more descriptive given that M&M comes after Results.

We have added the following sentences to the Results section:

L:118 “Following QC and relatedness filtering, the final datasets used for analyses consisted of 321 LR (178 cases and 143 controls), 256 GR (143 cases and 113 controls), 219 GSD (106 cases and 113 controls), and 235 WHWT (137 cases and 98 controls) with imputed marker sets of ~400-600K variants (**Supplementary Tables 1-2** and **Supplementary Fig. 1**).”

L:190 “The imputed datasets from 321 LR, 256 GR, 219 GSD, and 235 WHWT were used also in the XP-EHH analyses for detecting selection signatures in cases versus controls in each breed (**Supplementary Tables 1-2**).”

Comment#3

In line 134, you could check if there are epigenomic peaks in Barkbase for the candidate associated chr17 region.

#R: We have now intersected all effect variants as well as additional variants in the associated region on chr17 with BarkBase ATAC-seq data. The results from this are presented in **Supplementary Table S3** (effect variants) and **Tables S6-10**, and discussed in Results and Discussion.

We updated the following description to Materials and methods:

L:765 “The effect variants were intersected with BarkBase ATAC-seq data²⁶, and, for variants lifted to hg38, ENCODE cCREs²⁸⁻²⁹ (from UCSC Genome Browser) and GeneHancer³⁰ elements. LiftOver¹³¹ between genomes (canFam3.1, canFam4, and human hg38) was used to evaluate and compare functional and non-functional positions across assemblies.”

L:805 “Potential functionalities of variants were evaluated by SnpEff (v 4.3.t)³³ (canFam4 and a custom built database with the NCBI annotation of the reference), phyloP³², BarkBase ATAC-seq data²⁶, CpG annotation in canFam4, and, for variants lifted to hg38, ENCODE cCREs (from UCSC Genome Browser) and GeneHancer³⁰ elements.”

Comment#4

If there are some common variants that were detected by both BayesR and XPEHH?

#R: This is brought up in Discussion regarding chr26, we now modified the section to more clearly specify that there is an overlap between BayesR in LR with Selection results in WHWT (and GSD):

L:495 "One overlap between BayesR and XP-EHH regions across breeds was detected on chromosome 26. The effect variant chr26:34,371,008 in LR was located in an intron of *PCDH15* and the candidate region under selection on chromosome 26 in WHWT overlapped with a region under selection close to the cut-off in GSD ($-\log_{10}(p)$ XP-EHH =3.6). Top selection variants in both breeds were located close to or within the gene *PCDH15*, which has been associated with psychiatric diseases including bipolar disorder¹⁰¹. The gene *DKK1*, located 1.1Mb from *PCDH15*, has a documented role in skin pigmentation, thickness¹⁰², and inflammation¹⁰³."

We also added the following to the section after:

L:504: "Despite that there was no additional genomic overlap between BayesR and XP-EHH results apart from chromosome 26 mentioned above, the STRING enrichment analysis discovered several relevant pathways."

In addition, chr3 in LR was detected in both BayesR and XP-EHH and the deeper analyses across the selection region (as specified in Fig 4 and connected sections in Results/Discussion) indicate an overlap of the signals.

Comments#5

In line 191, you could check the variant effect for the *LRP1B* missense variant by SIFT, PolyPhen-2 or PROVEAN.

#R: this is now done and it is updated in Results:

L:204 "The putative impact of the exonic variant in *LRP1B* predicted in SnpEff (e.g., high, moderate or low) was moderate and SIFT^{35,36} predicted the substitution at amino acid position 42 to be tolerated with a score of 0.22 (SIFT score ranges from 0 to 1 and the amino acid substitution is predicted as damaging if the score is ≤ 0.05 , and tolerated if the score is > 0.05)."

Comment#5

In line 997, one column could be added to give information about the distance between the associated variant and the gene. The variant frequency in cases and controls should also be indicated in the table.

#R: We agree that this information is valuable. Since we use both canFam3.1 and canFam4 genome assemblies to extract all relevant information (given that the assemblies may provide slightly different information), the tables have a tendency to become complicated with too much information (or with columns with blank spaces). We included effect allele frequency in the breed in which the signal was detected. However, when it comes to the distance to the nearest gene (for intergenic variants) we settled for including the approx. distance between effect variant and nearest gene when specifically commenting on the candidate gene in Discussion.

Comment#6

The identified candidate genes could be searched in human GWAS database (<https://www.ebi.ac.uk/gwas/>) to see if they have some associations with human AD.

#R: We would like to thank the reviewer for this valuable suggestion. The search has generated additional information regarding the candidate genes presented in this study. We now report the findings in **Supplementary Tables 20-21** and discuss the genes in Discussion.

We specify the following in Materials & Methods:

L:751 "In addition, all genes located in BayesR regions (± 1 Mb from effect variants) and in XP-EHH regions (**Supplementary Tables 13-14**), were compared to associated genes from human GWAS of dermatitis, atopic eczema, eczema and psoriasis from the GWAS catalog¹³⁰ in order to detect gene overlaps between human skin disorders and canine AD."

Comment#7

Body weight genes like *TBC1B1* and *HMGA2* were detected in this study, but dog body size genes were strongly selected by human. *HMGA2* is strongly selected in small sized dogs. How could you tell if the detected signals are associated with AD rather than body size? Or is there a possibility to consider body size as a co-variate?

#R: *HMGA2* is located in the chr10 locus detected with bayesR of WHWT, hence it was not part of any candidate region under selection. Akey et al. (Tracking footprints of artificial selection in the dog genome. 2009. PNAS) showed that small-sized breeds compared to larger-sized breeds were highly differentiated near the *HMGA2* gene (Fst close to 1), but when they compared two small sized breeds (beagle and Jack Russel terrier) the Fst

was 0 at that same position. This indicates that small sized breeds are fixed for variants affecting the *HMGA2* gene resulting in a small size. Thus, it is most likely the selection for a small size in WHWT is rather fixed given that all dogs in the breed are small. We therefore do not think confounding by size by the *HMGA2* gene to our studied phenotype is likely.

The selection signal in LR overlaps with the *TBC1D1* gene. The hypothesis that LR belonging to one cluster are generally heavier/larger than the other can be supported by other studies of common type LR populations. Coat color is also known to vary across breed types (specifically chocolate color that is typically not found among gundogs). Also in our study, we see a division in LRs explained by breed type and supported by coat color. It could have been valuable to investigate body weight, however, it cannot be studied in detail within these datasets because individual body size measures are unavailable. There are also difficulties with body size measures when it comes to the time of measure and the effect by environment, i.e., is a heavier dog obese due to too much food/lack of exercise or is it due to a genetic predisposition for a large size? What we do propose within our study is that the inherited larger size dog has hitchhiking genetic risk factors for AD. But obesity due to environment is not, in our study, suggested to predispose for AD.

Reviewer #3 (Remarks to the Author):

In "Revealing risk factors for Canine Atopic Dermatitis using Bayesian model and selection signature analyses", the authors use an innovative approach to gain more insight into AD in dogs. Instead of using a traditional LMM GWAS, they suggest a Bayesian approach to account for the polygenic nature of the disease and to identify risk factor alleles rather than single high effect loci. I really like the complementary GWAS/ XP-EHH method as an attempt to investigate the hypothesis of potential hitch-hiking. Following the Bayesian test, which identified a major risk locus on Chr17 for LR, the authors conduct some sequencing to identify further variants associated with AD and give a comprehensive picture for this risk locus.

In general, it's an interesting study with sound methods and I'm impressed by the complexity/ the questions the authors have addressed to investigate the genetic basis of AD in dogs. There are two major points I'd like the authors to address:

First, readability is difficult at times (due to a lack of information/ explanation in Material & methods specifically and throughout the manuscript).

#R: We understand the point and have tried to improve the readability, in addition to other changes made as response to other points made by all three reviewers.

Second, grouping dogs into working/ show subpopulations based on PCA results is questionable and introduces bias in my opinion.

#R: We have addressed this matter. See specifics below.

Regarding the first point, I'm commenting on specific examples in the following, but I would encourage the authors to revise their manuscript in general with readability in mind.

• L492: "Swedish LR (...)" in title, but first sentence says "Dogs from LR breed (...)", which implies it's all LRs. This is very confusing throughout the whole manuscript, it appears you only grouped the Swedish LRs in show/ working, but when you look at the numbers (e.g. Figure 6) it is LRs from all countries (LRs add up to 322 here in the different groups, but according to L499-500 you could only group 33 and 51 LRs into the respective groups). Also Figure 6, there is no group for missing data (at least 19 LRs from Sweden with no information)? I later realised that you used the PCA to group all dogs into the subpopulations, but this is not properly explained anywhere. The same applies for GSDs.

R: We agree this has to be addressed and clarified. We have now added information regarding breed types for UK and Swiss dogs, and we have now also looked at coat color for LRs from all countries and this further support the split by breed type. We have substantially edited the entire section regarding LR in Materials and methods with the title: "Labrador retriever; breed types and coat color" starting on L571.

And similarly edited the section in Materials and methods about GSDs: "Swedish German shepherd kennel profiles and coat color" starting on L:595.

• L520: You describe sample-wise QC here (e.g. relatedness), but what about QC on genotype data? I understand that you filter for maf and genotype rate later, but there are also other parameters that are usually filtered for (e.g. Gene train score and signal intensity (ABRmean) in GenomeStudio).

R: The first step of genotype quality filtering was performed by the platform with standard/recommended protocol from Illumina. Genome studio is normally used to transfer the chip raw data (intensity of probe A and B) to genotypes. We started with a merged genotype dataset consisting of 167,211 SNPs and 1,152 dogs from four

breeds, not qc:ed on --maf, --geno nor --mind. QC was performed per breed (plink --geno 0.05 --mind 0.05 --maf 0.05) for the analyses (relatedness and PC estimation) on chip-SNP data. We performed QC prior to imputation on the whole dataset (--maf 0.0001, --geno 0.05, and --mind 0.05) and after imputation per breed (plink --geno 0.02 --mind 0.05 --maf 0.05).

We specify now in text:

L:627 "The CanineHD 230K BeadChip is an extension of the CanineHD 170K BeadChip and we started with a merged genotype dataset consisting of 167,211 SNPs and 1,152 dogs from four breeds."

L:632 "Quality control (QC) was performed per dog breed (plink --geno 0.05 --mind 0.05 --maf 0.05)."

Before Imputation:

L:653 "Quality parameters used in plink prior to imputation were --maf 0.0001, --geno 0.05, and --mind 0.05 and the dataset before imputation consisted of 1,152 dogs (347 LR, 294 GR, 231 GSD, and 280 WHWT) and 148,889 SNPs."

After Imputation:

L:669 "After imputation, the dataset was split into each breed and analyzed breed-wise (QC: plink --geno 0.02 --mind 0.05 --maf 0.05)."

- L524-528: It is good to give the exact specifiers used, but it is also helpful to explain what they are doing for people unfamiliar with the used tools. It is for example not very clear to me, if you have done the PCA on the genetic relationship matrices? Also, in which step do you remove samples (I assume "kin.thresh" filters for samples with relatedness < 0.125)?

R: An important point, we have clarified this by adding the following to:

L:633 "Genetic relationship was estimated using the KING method of moment for the identity-by-descent analysis¹¹⁹ in SNPRelate. Individuals with a kinship coefficient above 0.177 (~2nd degree relatedness) were removed to generate a dataset with highly related dogs excluded. PCs were estimated using pcair (part of the GENESIS R-package) with the following settings for 'snpgdsLDpruning': method='r', ld.threshold=0.7, slide.max.bp=250000, maf=0.05, missing.rate=0.05, and for 'pcairPartition' and 'pcair': kin.thresh=0.125, div.thresh=-0.125. In pcair, the PCs were estimated in a subset of individuals unrelated at the kinship coefficient threshold of 0.125, after that step PCs were projected on the more related individuals (*i.e.*, between 0.125-0.177). This was to avoid bias due to relatedness in the PC estimation."

- L531-551: I'd suggest to explain the steps in more detail (*i.e.* describe what the different specifiers mean, as mentioned before). It is for example not obvious to everyone that you prune the data in iii) with "--prune" and L549 poses the question where the first step of pruning was.

R: Sorry for the confusion, the --prune step in SHAPEIT2 has to do with the numbers of iterations within the Markov chain Monte Carlo (MCMC) step in the pre-phasing and is not part of the LD-pruning step of final markers. We added a few words to clarify this, but for more details it is better to refer to the SHAPEIT manual /homepage as all details cannot be specified within the manuscript.

L:659 "iii) the genotype data was pre-phased with the reference panel using SHAPEIT2 (with settings --effective-size 500, details on the Markov chain Monte Carlo iterations were --burn 10 --prune 10 --main 50, and threads -T 5)."

We had also missed to mention the first step of pruning in the text, it was previously only presented in Table S1. This has now been clarified with the following changes:

L668: "After imputation, the dataset was split into each dog breed and analyzed breed-wise. First, a QC step was applied (plink --geno 0.02 --mind 0.05 --maf 0.05). Next, we used plink to LD-prune the imputed datasets (--indep-pairwise 25 5 0.999) followed by adding all genotyped SNPs that were excluded in the pruning step. A second step of LD-pruning was performed in GR and LR using a stricter threshold (--indep-pairwise 50 5 0.99) since the total number of variants from the first step of LD-pruning exceeded fastPHASE (v. 1.4.8)¹²³ capacity (**Supplementary Table 2**)."

- Supp Table 1: In the text, Supp Table 2 is mentioned before Supp Table 1. As described later, some numbers are confusing. Would it maybe make more sense to restructure the data description?

R: In the Materials and Methods, the Table S2 was mentioned before Table S1. And in Results they were referred to together as Supplementary Tables 1-2. We have now changed the order of these two tables. We have also, following specific comments, rephrased parts of the data descriptions sections. We believe it is now more clearly described.

- L668: Why did you fit different factors for different breeds?

#R: We used a mixed model to fit fixed factors to the trait in each breed. We tested the PCs in each breed and only the significant PCs were included in the final model.

In Materials and methods, we specify the covariates for each breed with this text:

L:696 "Fixed effects were the first two (LR and WHWT) or three (GR and GSD) PCs (defined by "fitNullModel" in GENESIS to have significant ($p < 0.05$) effect on the trait). For GSD, the $-\log_{10}$ of IgA levels and $-\log_{10}$ of age in years at sampling were included as fixed effects in line with the described relationship between AD and low serum IgA levels in GSD¹²."

Results: I understand that the journal style requires the Results section before Material & methods. I would suggest to add some more information in the results section to help readers understanding what you have done.

#R: We agree. As part of other specific comments and additional adjustments with this in mind, this has become clearer.

- L118: Please explain how loci/ regions were determined (also in methods).

#R: Yes, this is now explained in:

Results:

L:124 "Variants with absolute effect size ≥ 0.0001 were defined as effect variants and AD-associated loci were regions harboring variants at $< 1\text{Mb}$ distance (Table 1, Supplementary Table 3)."

Materials & methods:

L:704 "Effect variants separated by $> 1\text{Mb}$ were considered to belong to separate associated loci and the effect variant with the highest absolute effect size was extracted to represent the associated locus."

- L120: Why this threshold of effect size for effect loci?

#R: The value of 0.0001 was taken from the lowest effect size distribution and we applied this for all breeds to generate comparable results. The setting of thresholds is not easily defined. In a previous study in dogs, the top 50 variants were reported as the effect variants (Baker, L. A. et al. *Biologically enhanced genome-wide association study provides further evidence for candidate loci and discovers novel loci that influence risk of anterior*. 2021). This could be an option for us as well, but, as brought up in the Discussion section, it may also yield loci with negligible importance or false positives. Another issue is whether or not to use the same effect size cutoff for the four different breeds. There are arguments for and against it. Arguing for it is to get a good comparison in terms of plotting and interpreting results. In Stephen and Baling 2009 (Bayesian statistical methods for genetic association studies. *Nat. Rev. Genet.*), they propose that "Bayesian methods compute measures of evidence that can be directly compared among SNPs within and across studies". Arguing against it is that populations likely have risk factors with different effect sizes and the modelling in the various breed populations with different number of dogs and variants will generate diverse results. Since we wanted to combine the results from AD mapping in four breeds, we sought to be able to compare and combine the outcome in the most optimal way and aiming to decrease the risk of false positives a rather strict cutoff was applied. However, since we in three of the studied breeds only define one or two associated loci in the BayesR analysis, it indicates a possibly too strict cut-off chosen because multiple risk factors were expected. With this in mind, we have now as a complement, added a list of the top 50 effect variants for each of the breed (**Supplementary Table 22**) in Discussion. This less strict cutoff generated additional loci and additional support to the top loci by adding variants. In total, this cutoff yielded 50 associated regions (13 in LR, 19 in GR, 9 in GSD, and 0 in WHWT). One could go back to results at a later stage and validate these lower effect size variants for these breeds. However, within this publication we decided to use the more stringent cutoff to emphasize the evaluation of regions with variants of the slightly higher effect sizes in detail.

To further clarify this choice of threshold, we have now added the following sentence to Materials & Methods:

L:700 "To determine a rational cutoff for defining effect variants throughout all four populations of breeds, we chose the value of 0.0001 from the lowest effect size distribution and applied this for all breeds to generate comparable results."

And we added the following to discussion including Supplementary Table 22:

L:311 "In three breeds, only one or two associated loci were identified in the BayesR analysis, which indicates a too strict cut-off chosen because multiple risk factors were expected. However, lower effect sizes increases the risk of false positives²¹. In the study of ACL in LR using BayesR, the top 50 effect variants were presented²³. That approach in our study would result in additional markers in the already defined associated loci but also additional loci harboring variants with lower effect sizes, totaling in 50 associated loci (**Supplementary Table 22**). As a

follow-up study, these loci could be further investigated or increased sample sizes could define additional lower effect variants of relevance to canine AD with higher certainty.”

- L129: Principal component 1, explain here what PC1 is (from PCA on genotypes), what does PC1 characterise?

#R: PC1 is the first dimension in the relationship plot (PCA plot) of the specific population (in our case, the individual dog breed). The PC calculation captures trends in the relationship patterns in a population and the first PC is what explains most of the difference. In the calculation made in the section around L129 (of the first submitted manuscript), we investigated the risk of being an AD case was affected by PC1. In that case, one can draw the conclusion that the risk for AD is correlated to subpopulations or substructure explained by the PC1. PC1 for some breeds can be (partly) explained by a breed type division visualized as subpopulations and in some breeds, there is no clear division explained by PC1. We added to a brief explanation to the text:

L:136 “Principal component (PC)1, which captures the first dimension in the relationship matrix (PCA plot, **Supplementary Fig. 1**), contributed 9.3% to the total AD variance in the risk index model and 5.2% when modeling loci separately (**Supplementary Tables 4**).”

Based on information from the Swedish kennel club (SKK) we found that the first PC likely characterizes breed type differences in LR and GSD. This was further supported by coat color for all LR and GSD in Sweden, and breed type information of LR from the other countries represented. See more details below and updated **Supplementary Figures 3-4**.

Moreover, WHWT show a split into subpopulations based on country of origin, with the Swedish WHWT clustering more tightly together and partly overlapping with the US population, which appears more diverse (**Supp Fig 1d**). Generally, it is known that there are two breed types also of GR, similarly to LR, and from our data it seems to be a few dogs in the right-hand cluster (**Supp. Fig 1b**), possibly these are gundogs. However, the results did not give us any reason to follow this up.

Generally, Tables and Figures need more comprehensive explanations and legends

#R: We have tried to explain ourselves better in both legends and text.

- Supp Table 5: “Residual” missing
- Why Supp Table 5 and 6 separate?

#R: Good point, Supp Table 4 includes AD-associated loci in LR and Supp Table 5 risk index in LR. We combined these into one table: Supp Table 4 (and subsequently renamed the following sub-tables) and added the missing “residuals”. We did not combine it with Supp Table 6, just to make the Tables readable, as the number of AD-associated loci in LR are quite many.

- Supp Table 7: Explain headers
- Table 2: Are statistics given as average for region?

#R: No, all columns following column 4 are statistics for the top variants per locus, not for the whole region. We clarified this by removing the column Chr (since the chr is clear when the region is specified) and with a foot note to Table

2: “Column 2-4 refer to the candidate region under selection.” and “All columns following “Top variant ID” refer to the top variant.”

- Supp Fig 2: XP-EHH between which populations? Clarify that it's cases vs. controls

R: We rephrased to “Whole genome selection analysis presenting XP-EHH values from comparing the case with the control populations of LR (a), GR (b), GSD (c), and WHWT (d).”

- Supp Fig3: Use show/ working instead of “field” for consistency

R: This is now updated with the consistent terminology (see details below). We now use “gundogs” and “common type” when referring to LR breed types both the main text and in the figures throughout the manuscript.

- Fig4/L1045: This is only Swedish LR? Was the PCA only performed for Swedish LR? XP-EHH only for Swedish dogs?

R: We agree that this was not entirely clear. PCAs were performed on the whole dataset of LRs, however with the following details (as also explained above): In PCAir the PCs were estimated in a subset of individuals unrelated at the kinship coefficient threshold of 0.125, after that step PCs were projected on the more related individuals (*i.e.*, between 0.125-0.177). This is to avoid bias due to relatedness in the PC estimation. Thus, they

were not split by country in the PCA estimation but we utilized PCAir to improve the PC estimation in a related dataset such as this.

We have now updated **Fig 4a** to include only genotype and not breed type information. Instead, the breed type support as well as coat color information are presented only in the supplementary information and further discussed in the manuscript.

The new figure **4a**:

To clarify we have now updated the text to:

Figure 4 legend L:1249 “The top XP-EHH region in LR was located on chromosome 3. Information about Swedish LR kennels, from UK owner questionnaires, and information regarding Switzerland working dogs suggest that LR of gundog type are present in the low PC1 cluster whereas the common type LR are represented in the high PC1 cluster. This is further supported by the chocolate coat color only found among the common type LR (**Supplementary Fig. 3**). The low PC1 cluster is subsequently referred to the gundog subpopulation, whereas the common type LR are in the high PC1 cluster.”

We also clarified in the Results that XP-EHH was performed on the entire datasets for each breed:
L:190 “The imputed datasets from 321 LR, 256 GR, 219 GSD, and 235 WHWT were also used in the XP-EHH analyses for detecting selection signatures in cases versus controls in each breed (Supplementary Tables 1-2).”

• Supp Table 4 and 5: Why no PC3 fitted?

R: In Materials and methods we specify the covariates for each breed with this text “Fixed effects were the first two (LR and WHWT) or three (GR and GSD) PCs (defined by “fitNullModel ” in GENESIS to have significant ($p < 0.05$) effect on the trait). For GSD, the $-\log_{10}$ of IgA levels and $-\log_{10}$ of age in years at sampling were included as fixed effects in line with the described relationship between AD and low serum IgA levels in GSD¹⁰.”

Thus, we use only the PCs with significant effect on the AD phenotype in each breed. This is why it differs between breeds.

• Supp Table 6: Why two sub-tables for WHWT? Why no PC3 fitted for WHWT? Explain whole table better (e.g. that loci fitted are risk loci)

R: See explanation above. We also added a note to Supp table 5 (previous Supp table 6):

“Footnote: Fixed effects were defined by “fitNullModel ” in GENESIS to have significant ($p < 0.05$) effect on the trait.”

• L481 and Supp Table2: 67 and 57 in text vs. 62 and 48 cases in UK LR and GR in Table (which number is correct?). The same non-matching numbers for controls. I assume it is number of dogs with phenotype and genotype in Table, whereas it is only dogs with phenotypes in text. Maybe make this clear in the Table and also mention this in text when you make the reference to Supp Table 2 (491)?

R: Yes, that is correct, we understand now that it was confusing. We have clarified:

L:559 “In total, 67 LR and 57 GR cases respectively, were included with well-classified phenotypes from the UK sample set.”

L:564 “In total, 48 LR and 68 GR were defined as well-classified controls for the current study. The majority of included cases and controls had available genotypes and were used in subsequent analyses.”

L:568 “An overview of sample distribution across countries is presented in **Supplementary Table 1**, including only the samples with genotype data available.”

- Table 3: “XP-EHH” instead of “Selection”. How was function “immunity” or “epidermis” assigned?

R: We agree that it should be used in concordance to BayesR. We now changed Selection region to XP-EHH region (except when describing the XP-EHH region as candidate regions under selection) throughout the manuscript including in Table 3.

The functions were assigned the broad terms immunity and epidermis based on references, these are presented in the Discussion section. See the brief summary here (in purple generalizing to belong to the broader term “Immunity”, in red to “Epidermis”):

CLNK (IgE)

ECM1 (epidermal diff.), OTUD7B (B-cell), VPS45 (neutrophils), BCL9 (B-cell)

DKK1 (epidermis)

ARL14 (MHCII, dendritic cells), KPNA2 (epidermal diff.)

ITGA4 (leukocytes, T-cells)

EPHA4 (epidermis)

ACVR2B (allergy, skin homeostasis)

ABCA9 (monocytes)

IRAK3 (monocytes, epithelial cells, TLRs)

KLF3 & MSX1 (epithelial diff.), CLNK (IgE), TLR (T- and B-cell)

KYNU (epidermis & infiltrating immune cells)

FCER2 (B-cell, IgE)

SMOC2 (epidermis, cell-matrix), THBS2 (epidermis, cell-matrix)

DKK1 (epidermis)

- Supp Figure 1: Needs more explanation in the figure legend (it is PC scores, why relationship matrices?)

R: We rephrased to PCA plot to avoid confusion.

One can argue that PCA is better at showing global structure, i.e., genetic distance/relationship between breeds/populations whereas MDS is more precise when studying the distance between individuals (pairwise estimates). However, PCA is also a relationship matrix and is already a part of our analyses pipeline and in practice equivalent to MDS.

We changed the legend to **Supp Fig 1**:

“**Supplementary Fig. 1** PCA plots displaying the genetic relationship color coded with country of origin for each breed, LR (a), GR (b), GSD (c), and WHWT (d).”

- Figure 4a/ Supp Figure 3: it's both PCA on Swedish LR, why are there differences in the plot? Again, I now understand that Figure 4 is all dogs and Supp Figure 3 is only Swedish dogs, but as a reader you think you only have dog type information on Swedish LRs (because you don't describe your PCA approach to group dogs), which leads to a lot of confusion throughout the manuscript.

R: Sorry for the confusion. We agree that this was not clearly stated. The PCs were estimated on all LR dogs and used also for the supp figure plot, even if only Swedish dogs were kept in the plot. We have now updated the **Figure S3** to include all LRs and we have also utilized additional information about coat color for all LRs and breed types from UK and Switzerland.

Breed type information based on registered gundog LR kennels for a large proportion of dogs was only available for the Swedish LRs. However, the additional information from UK and Switzerland have provided us with further support for the breed types. See details below.

Regarding the second point, I had initial problems to understand how you only have dog type information for some Swedish LRs and some GSDs, but perform comparative analyses grouping all LRs and GSDs into show/working. As far as I am aware, you only explain in the legend of Figure 5 that you use the PCA results and where the Swedish LRs and GSDs with known dog types cluster to decide that scores for PC1 discriminate between dog types. Clusters can be attributable to other factors (e.g. country in case of LR, coat colour etc.) and in my eyes you don't have a strong case for a confounding between dog type and AD based on your assumptions/methods. I would suggest to a) only include LRs and GSDs for which the dog type is known to compare risk alleles or cases/controls in subpopulations or b) group dogs into high vs. low PC1 scores and discuss later what PC1 might be associated with.

R: We understand the point made here and want to avoid that we draw this conclusion on the confounding statement at a too early stage. We have addressed this issue in detail, obtaining color information, knowledge about the Swiss cohort and specific questionnaire information from UK. We used these different levels of support for the subpopulation formations described by PC1. See details in the response to the next question.

• Figure 5/ L216: If I understand this correctly, you group all LR into working/ show based on the PCA scores for Swedish LR? I am not convinced that this is a suitable approach, as stated above, PC1 could also capture variation explained by "country". Using your assumption, you'd conclude that all NE and Suisse dogs are show dogs, is this likely?

R: Regarding the Swiss cohort (including also Dutch LR): At the time of collection (~2010) working / show lines was not discussed. Dog sports were less present 12-15 years ago in Switzerland and due to the geography hunting is more mountain-oriented, as opposed to the classical English gundogs. Labradors used for work in Switzerland are primarily not of field trials/hunting as this is limited there. Instead, Labradors are used for example as snow avalanche rescue dogs, guide dogs for the blind, and that the will to please / retrieve is used for different types of work. Some guide dogs for the blind were included in the Swiss cohort but this type of working dogs is generally not from a gundog breeding background. **Plot 1** shows all LR and their country of origin (same as **Supplementary Fig. 1a**), to use as comparison with the following plots. For the Swiss cohort, there were 11 guide dogs for the blind (green color in plot) and one Swiss LR was a police dog and end up in the low PC1 cluster (**Plot 2**). The phenotypes of the collected dogs were, as author Petra Roosje (who was sampling the majority of dogs of the Swiss cohort) recall, rather "moderate", i.e., lacking extremes of working/ field trial dogs or show type LR. According to the Dutch LR club, they initially had more working type dogs but switched from the 1980's towards the show type including increased popularity of the chocolate color.

Plot 1: All LR origin (black= Netherlands, red= Switzerland, green=Sweden, blue = UK).

Plot 2 (green = Swiss guide dogs for the blind, red= police dog, the rest is grey)

For the UK LR, we extracted information from questionnaires on whether the dog, according to the owner, was either a “gundog”, “show dog”, or “working dog”, see **plot 3**. Note, the working dog is also of chocolate color positioned in the “Swiss dog cluster”, see **plot 1 (country of origin)** for comparison.

Plot 3. UK dogs (gundog= red, show dog= green, both gundog/show dog = purple, working dog = turquoise)

Looking at the breed type of Swedish LR in **plot 4** we see that LR from kennels registered as gundog-kennels are more common in the low PC1 cluster compared to the rest of the Swedish LR that are from LR kennels not listed as gundog kennels. This is in concordance to the UK gundogs.

Plot 4: Swedish LR breed type (gundog= red, Swedish LR not listed as gundog = black, unknown = grey)

Next, we extracted coat color for all LR. Coat color in the Swiss LR is shown in **Plot 5**, and 20% of LRs in the Swiss cohort were chocolate-colored (23% of the Swiss dogs and 14% of the LR from Netherlands). Color information was extracted from questionnaire data from the UK dogs and 24% were of chocolate color and they are only found in the high PC1 cluster (**plot 6**). Also, in the Swedish LR, chocolate color dogs positioned among the high PC1 values (**plot 7**) in concordance to the UK chocolate-colored dogs. A lower proportion of the Swedish LR (9.5%) were of chocolate color compared with Swiss and UK LR. Proportions were calculated from the dogs with known color. See **Table 1** for the number of dogs per color and country of origin. See **plot 8** for all breed type and color information combined into a final figure (**Supplementary Fig. 3**).

Plot 5 Swiss LR coat color (black = black coat color, red = yellow coat color, green = chocolate color, non-Swiss dogs in grey)

Plot 6 UK LR coat color (black = black coat color, red = yellow coat color, green = chocolate color, non-UK dogs in grey)

Plot 7. Swedish LR coat color (black = black coat color, red = yellow, green = chocolate, NA= grey)

Plot 8: All LR breed type (a) and coat color (b) also Supplementary Fig. 3

Table 1. LR coat color per country

	Finland	France	Netherlands	Sweden	Switzerland	UK
Black	0	0	26	63	37	47
Yellow	1	0	9	21	23	24
Chocolate	0	0	5	8	14	17
NA	0	1	0	10	6	9

Finally, we wanted to verify that the selected allele was not *only* present in the chocolate color dogs (and thereby possibly linked to coat color). We can confirm that many chocolate color dogs have the allele C, but not all, and that many black and yellow dogs also carry the C allele (**Table 2**). Therefore, we conclude the selected genotype is not linked to coat color.

Table 2. Genotype of the top selection variant on chromosome 3, C is the selected allele.

Color	T/T	C/T	C/C
Black	19	57	97
Yellow	8	28	42
Chocolate	3	11	30
unknown	6	7	13

Changes made to the manuscript:

In Results:

L:235 “When dividing the dogs into subpopulations by setting the cutoff at PC1 = -0.05 (gundogs PC1 < -0.05 and common type PC1 > -0.05), it became clear that a larger proportion of common type cases was homozygous CCG, and the CCG allele frequency was also associated with AD in the common type ($p=2.8 \times 10^{-5}$; chisq.test; Fig. 6a).”

When we study the distribution of the selected allele across the PCA plot (**Plot 9** also **Fig. 4a**), we note that the selected allele C in homozygous form (grey) or heterozygous (yellow) is more dominant among the high PC1 dogs, and that T/T (purple) is more common among the low PC dogs.

Plot 9. Chr3_sel genotype in LR. T/T purple, C/T yellow, C/C grey.

From all this we can support a breed division by breed type. We have also updated our terminology as the working vs show type is more complex than what we originally referred to, specially after addressing the information about working dogs from Switzerland. Consequently, we have changed our terminology throughout the manuscript to refer to **gundog** and **common type**, to better describe what we are really referring to. Common type LR is a wider term including both show type and working type from Swiss breeding background and previously used in other publications referring to breed types in LR (Sundman, A.-S., Johnsson, M., Wright, D. & Jensen, P. *Similar recent selection criteria associated with different behavioural effects in two dog breeds. Genes Brain Behav.* 15, 750–756 (2016)). Interestingly, a study of behavior differences in LR split by coat color also concluded that black dogs showed a higher fetching tendency than chocolate dogs and chocolate dogs also demonstrated a lower trainability and higher incidence of unusual behavior than black or yellow dogs (Lofgren, S. E. et al. *Management and personality in Labrador Retriever dogs. Appl. Anim. Behav. Sci.* 156, 44–53 (2014)). This supports the fact that chocolate-colored dogs are generally not detected among gundogs.

Throughout the manuscript we now refer to the gundog and the common type LR and specify the terminology of common type vs. gundog with support from our data as well as from previous publications.

In Materials and Methods:

L:571 See the whole section with the heading “Labrador retriever; breed types and coat color”

In Results:

L:209 “There is a substructure in the relationship matrix of LR (Fig. 4a and Supplementary Fig. 1a) and by utilizing information from Swedish LR kennels, questionnaires from UK, and coat color information from all LR, we concluded that the PC1 likely captures a breed type division caused by selection for a gundog versus a common type LR. Gundogs were more often in the low PC1 cluster, subsequently referred to as the gundog subpopulation, and the cluster with high PC1 values was considered as the common type subpopulation (Supplementary Fig. 3).

Fig 4a now includes only coding for genotype and not for breed type in the PCA plot. Additional information is instead to be found in **Supplementary fig. 3**.

Fig4 legend: L:1247 “The top XP-EHH region in LR was located on chromosome 3. Information about Swedish LR kennels, from UK owner questionnaires, and information regarding Switzerland working dogs suggest that LR of gundog type are present in the low PC1 cluster whereas the common type LR are represented in the high PC1 cluster. This is further supported by the chocolate coat color only found among the common type LR (Supplementary Fig. 3). The low PC1 cluster is subsequently referred to the gundog subpopulation, whereas the common type LR are in the high PC1 cluster.”

Fig 6 was updated to refer to “Gundog type” and “common type” for LR.

Supplementary Fig 3 legend:

“LR from Swedish kennels that were listed as gundog type kennels are marked blue in the relationship matrix. The list of gundog type kennels included data from kennels with puppies born 2006-2021. (Three Swedish LR in the low PC1 cluster were called as common types but these are from gundog kennels active before 2006, i.e., not included in the current gundog kennel list.) Furthermore, seven UK LR were defined as gundogs (blue), two as

“show” dogs (green), one was both gundog/show dog, and one working dog (turquoise) based on owner questionnaires. One police dog (black) and eleven guide dogs for the blind (purple) were defined in the Swiss cohort (a). The coat color was extracted from all dogs except 26 showing that the chocolate color (green) does not exist in the “gundog” cluster as expected (b).”

In Discussion:

L:410 “Interestingly, a study of behavior differences in LR split by color concluded that black LR showed a higher fetching tendency than chocolate colored LR, and that chocolate dogs demonstrated a lower trainability than black or yellow dogs⁶⁸. This supports that gundogs generally are not chocolate-colored. Our results similarly point to a split into two LR subpopulations and ... “

• Figure 6: You report information for 217 GSDs (why does it say 219 dogs in Supp Table 2?), and I assume you have again used PCA to split dogs into show/ work? In GSDs, population structure can also be confounded with coat colour (working dogs darker phenotype).

R: Two GSD control dogs did not have a genotype at this position, therefore the number 217 instead of 219. We now added that information to the **Fig 6** legend.

We address the question regarding confounding with coat color in GSD below.

• L1046: “thus we can conclude that the division formed in the relationship matrix can be attributed to working type LR in the low PC1 cluster and show type LR in the high PC1 cluster”: But you only have this information for Swedish dogs, i.e. a high number of “points” must be “NA” (it is not very clear from the points size). When you look at Supp Fig 1, you could conclude that high scores in PC1 are attributable to dogs from NE and Switzerland, hence indicating a confounding with country rather than with working/ show dog?

R: See previous answer for details regarding LR breed types. Both UK and Swedish dogs clearly show an overlap between countries following the breed type rather than being split by country. This is also confirmed with the chocolate color dogs, which primarily exist among show/common type dogs. NE and Switzerland do group among the common type rather than the gundogs, where Switzerland LR seem to form a separated group overlapping with common type/show type LR rather than gundogs.

• L359: Interesting point. However, as I mention earlier, the two subpopulations indicated by PC1 might be explained by attributes other than dog type (especially since you don’t have information for all dogs under analysis). Label groups high vs. low scores for PC1 instead of “working vs. show” and try to disentangle what factor might be confounded with PC1 (e.g. dog type, coat colour, country in LR). You suggest here that body weight is what differentiates the two subpopulations, if that’s the case one conclusion could also be that there is a difference between countries, e.g. that NE and Switzerland select for heavier dogs.

R: The selection is likely towards show/common type dogs, i.e., a heavier dog, in Switzerland and the Netherlands. However, some of these dogs are used in work, but that this type of work and this type of working dog is different from the gundogs that are primarily present in Sweden and UK. We see an overlap of gundogs from the UK and Sweden in the low PC1 cluster and the common type (or show type) LR in the high PC1 cluster, which indicates that the division is rather by breeding goals than that they are separated by country.

• L400: Again, interesting point. I think it would be more significant if comparison happened only between GSDs for which working/ show status was known (excluding all others instead of extrapolating them based on PCA results).

R: First, we have to remember that working vs show status is an estimate based on kennels and their achievements in working vs show events. It does however seem to be a good approximation for describing PC1 variance. We now also address the coat colors in the GSD, knowing beforehand that the grey coat color is a typical working type color and the more standard GSD coat color; brown dog with a black saddle, are present in both types.

Colors could be extracted from 192 dogs whereas breed type based on kennel merits was estimated in 121 dogs. See the owner-recorded colors (from the Swedish kennel club) in **Table 3**. The colors marked in grey were grouped as the color-class “grey/black”, and the colors marked in brown included colors with any brown/red/yellow markings defined as color-class “brown and black”.

Table 3. GSD coat colors

	Show type work prop <0.5	Work type work prop >0.5	NA
grey	1	22	14
dark grey	0	0	1
black with grey markings	0	0	2
black	0	6	1
black with grey-brown markings	0	1	0
grey with brown markings	1	5	1
grey with yellow markings	0	2	0
grey with red markings	0	0	1
black with brown markings	23	41	38
black with yellow markings	3	10	6
black with yellow-brown marking	0	1	2
black with light markings	0	1	2
black with red markings	2	2	3
NA			27
Sum	30	91	

In the PCA plot (Plot 10 also Supplementary Fig. 4), it becomes clear that the grey/black color is more common among the low PC1 dogs, however it is not the only color in that cluster.

Plot 10. GSD work/show proportion (a) and coat color (b).

When addressing the question if the selected allele is linked to coat color, we found that the selected allele exists in both color classes (Table 4). In the subset with information from both breed type and coat color (N=121), we used t-tests to i) compare the groups of color-class vs. genotype ($p=4.0 \times 10^{-8}$) and breed type vs. genotype (2.5×10^{-8}). ii) to estimate the difference in PC1 values by color-class ($p=3.0 \times 10^{-80}$) and breed type ($p=2.6 \times 10^{-79}$). ANOVA indicated that the phenotypic variance explained by PC1 was 35% for color-class and 26% for breed type. In summary, both these estimates show a clear difference in coat color and breed type distribution across the PC1, which indicates there is a split by breed type and that one coat color is more common in one of the subpopulations/ breed types. Breed type explained 10.5% of the AD variance and color-class explained 6.0%. When modelling both breed type and color in the same linear model, color explained 6.0% and breed type 7.1% of AD (and only breed type was significant $p < 0.05$). This indicates that part of the information from breed type can be extracted from the color information. However, breed type is still more informative when explaining AD. The numbers of AD cases and controls with the two different coat color classes are shown in Table 5. We draw the conclusion that the color is a proxy for breed type rather than being linked to the selection locus and risk for AD.

Table 4. Selected allele and coat color in GSD. T is the selected allele

	C/C	C/T	T/T	NA
Grey or black	6	17	24	0
Black+brown	29	58	56	2

Table 5. Coat color in AD cases and controls

	AD cases	Controls
Grey/black	14	33
Black & brown	78	67

We now support the breed type division by adding coat color information. **Supplementary Fig. 4** now includes one more panel of coat color class (i.e., **plot 10**). We also changed panel **Fig 5a** to only include genotype information.

New fig 5a:

The following updates was made to the manuscript:

In Materials and Methods:

See the entire section: "Swedish German shepherd kennel profiles and coat color"

In Results:

L:240 "A division in the GSD breed into two subpopulations can be visualized in the PCA plot. We assigned the subpopulations to working type (PC1<0) and show type (PC1>0) GSD based on the following information: GSD coming from kennels with a higher proportion of dogs with working merits compared to show merits were more common in the cluster with low PC1 values and vice versa, and GSD with black or grey coat color (typically observed among working type GSD) were almost exclusively present in the low PC1 cluster (**Fig. 5a** and **Supplementary Fig. 4**)."

Fig. 5 legend:

L:1273 "Dogs from GSD kennels with a higher proportion of working compared to show merits and with grey or black coat color were more frequent in the low PC1 cluster whereas the opposite was seen for dogs in the high PC1 cluster (**Supplementary Fig. 4**). The subpopulations were subsequently referred to as the working (low PC1) and show (high PC1) type subpopulations. Dogs homozygous for the T allele were more frequent in the working subpopulation and the pictures illustrate the common grey color in a working type GSD and a show type with the typical GSD color pattern of brown with black saddle and a sloping cross (**a**)."

In Discussion:

L:444 "Selection for different breed types in GSD was also evident in our data"

Other points:

Apologies that the comments are not entirely in order, but that's the order I read through the manuscript.

L62-63: I'd like to positively highlight the author's efforts in being very precise with the assembly version to enable comparisons between studies.

Please consider rephrasing L68-83: You specify limitations (not assumptions) of GWAS, which can be solved with appropriate data (e.g. small effect sizes can be detected when sample size is large enough, relatedness can be accounted for by fitting relationship matrix/ remove highly related samples), WGS is not an approach (you can do a GWAS with WGS, also, by only sequencing a couple of samples you won't be able to identify rare variants):

- L68-71: Maybe reconsider your description here? I think what you're trying to say is that in LMM, you test for associations between a single marker and the trait and pick up only markers with high effect size, whereas with a Bayesian approach, you test the joint effect of a set of markers (predict the risk for AD based on these loci).
- L76: You can do a GWAS with WGS data? Or is what you mean that instead of SNP array genotyping WGS is able to identify rare variants?
- L77: what do you mean with "evolutionary constraint variants"?

R: Good points. We could update to the following: First of all, a good genome coverage is crucial, which can be gained by whole genome sequencing (WGS), and also, detecting rare variants require large sample sizes. Furthermore, functional evaluation of rare protein coding variants, or evolutionary constraint variants (i.e., variants conserved across many species) and their effect on gene expression in relevant tissues should be considered.

However, after revising the manuscript we decided to exclude this section about rare variants from the introduction to get more quickly to the scope of this paper.

Regarding constraint sites we did update this sentence in Results:

L:195 "We investigated potential functionality of selection variants, i.e., variants with $-\log_{10}(p)$ XP-EHH ≥ 4.0 (N=1,471), by extracting the phyloP³² scores (Supplementary Table 11). We found that 12 selection variants were positioned at constraint sites (phyloP > 2.56 ; i.e., showing a high level of conservation across 240 species and thereby a likely functional position) on chromosomes 3, 10, 19, and 32."

Regarding Limitations vs Assumption, we rephrased to:

L:67 "A limitation in a traditional GWAS (e.g., linear mixed model, LMM) of a complex trait is that it primarily captures a single or a few risk factors with high effect size when, instead, multiple risk factors with effects ranging from small to moderate are expected to jointly influence the development of a complex trait."

- L78-83: Use genomic relationship matrix? As long as you have genotype data you can calculate relationship between samples and include it in your model. (see here <https://www.nature.com/articles/ng.3190> "Linear mixed models are emerging as the method of choice for association testing in genome-wide association studies (GWAS) because they account for both population stratification and cryptic relatedness and achieve increased statistical power by jointly modeling all genotyped markers")

R: It sounds good to be able to include relationship matrix as random effects in a Bayesian mixture model. In our analysis, we used PCs to correct for relatedness which is common practice in traditional GWAS. However, one can argue that Bayesian models corrects for relatedness given that it models variants simultaneously (Yu J, Pressoir G, Briggs WH, Vroh Bi I, Yamasaki M, Doebley JF, et al. A unified mixed-model method for association mapping that accounts for multiple levels of relatedness. Nat Genet. 2006;38:203–8.) and possibly neither PCs nor genetic relationship matrices are necessary. But one must also remember that even if PCs or relationship matrix is used to correct for population stratification and cryptic relatedness, it is important to exclude highly related individuals. It may be possible to correct for relatedness to a certain extent but not fully. In dog populations a major problem is the high degree of relatedness, breeds are by definition isolated populations (due to closed stud books) and most breeds have a history of population bottlenecks and a limited set of breeding animals (i.e., low effective population size). They cannot be compared with large and diverse human sample sets and even if the optimal level of unrelatedness mostly cannot be achieved in dogs, we have to be careful with including highly related individuals as they introduce bias in the results.

L99: Remove “,”

R: done

L462: How where the dogs chosen/ invited to participate? You explain in the Reporting Summary it's dogs (or better their owners) seeking health care, which I would include in Material and methods

We updated the following in Materials and Methods:

L:614 " The Swiss cohort included samples from dogs collected in Switzerland, Netherlands, Finland, Germany, and France. Dogs were recruited to the project as their owners visited the veterinary clinic to seek health care for AD (cases) or unrelated problems (controls), or were recruited as healthy controls followed by a visit to the veterinary clinic to leave blood samples. This applies to all countries except UK. Saliva samples from the UK (LR

and GR) were collected by owners of the dogs and posted to the research team as part of the questionnaire study⁶⁶, and...”

L489: What about phenotyping for remaining countries (e.g. Germany, Netherlands, France)?

R: They were part of the Swiss *cohort*, and all info for the Swiss cohort applies to the samples from Germany, Netherlands, and France.

L499: According to Supp Table 2, it's 102 LRs with Swedish origin.

R: The correct number is 102, and 32 (instead of 33) had kennel names included in the list of gundog kennels. This is now changed in:

L:579 "Of the 102 LR in our dataset with Swedish origin, 32 had kennel names included in the list of gundog kennels,..."

L525: Give reference for pcair

R: PCAir is part of the GENESIS R-package, which has included version and reference a few lines above.

To clarify we changed to:

L:636 "PCs were estimated using pcair (part of the GENESIS R-package) with the following settings for 'snpgdsLDpruning': method='r', ld.threshold=0.7, slide.max.bp=250000, maf=0.05, missing.rate=0.05, and for 'pcairPartition' and 'pcair': kin.thresh=0.125, div.thresh=-0.125."

L534: Mention that you impute to WGS

R: We do not impute the data to become whole genome sequences, so we believe it would be problematic to write WGS. We did use variants called from WGS to create the reference panel and from this we imputed our SNP-array genotype data to become the "imputed dataset".

L539: Is there something missing here "breed-dogs"?

R: We see now that this was unclear. We meant that pure-bred dogs were extracted from the canid dataset, which also included village dogs, wolf and other canids. We rephrased:

L:646 "We imputed the genotyped dataset using a reference panel of purebred dogs (435 individuals) extracted from a publicly dataset including wolf and other canids⁹¹."

Supp Table S1 and S2: Total number of LR differs (321 vs 322).

R: The mistake has been corrected now. The number of Swiss controls should be 42 instead of 43 in **Supplementary table 1**.

L568: Change to "belongs"

R: Done

L578: Why was this threshold chosen?

R: As specified in a previous response we have now performed the following update:

L:701 "To determine a rational cutoff for defining effect variants throughout all four populations of breeds, we chose the value of 0.0001 from the lowest effect size distribution and applied this for all breeds to generate comparable results. Therefore, we regarded variants with mean absolute effect size larger or equal to 1.00×10^{-3} as "effect variants".

And we discuss adding a comparison to a less strict cutoff. ...:

L:313 "However, lower effect sizes increases the risk of false positives²¹. In the study of ACL in LR using BayesR, the top 50 effect variants were presented²³. That approach in our study would result in additional markers in the already defined associated loci but also identify additional loci harboring variants with lower effect sizes, totaling in 50 associated loci (**Supplementary Table 22**). As a follow-up study, these loci could be further investigated or increased sample sizes could define additional lower effect variants of relevance to canine AD with higher certainty."

L579: Clarify what you consider as region.

R: Effect variants more than 1Mb apart were considered separate regions. This is clarified in:

Materials and Methods:

L:704 “Effect variants separated by >1Mb were considered to belong to separate associated loci and the effect variant with the highest absolute effect size for each locus was extracted to represent the associated locus.”

Results:

L:124 “Variants with absolute effect size ≥ 0.0001 were defined as effect variants and AD-associated loci were regions harboring effect variants at <1Mb distance (**Table 1** and **Supplementary Table 3**).”

L591: How does the function define regions of selection?

R: We have now specified the settings used by the function: (settings were threshold = 4, pval = TRUE, window_size = 1E6, overlap = 1E5, min_n_extr_mrk = 2) as a response to a previous comment. This means that in order to be considered a candidate region, there has to be at least 2 extreme markers where extreme = having a $-\log_{10}(p) \Rightarrow 4$ within a window of 1Mb. This scanning is performed in a window-based manner with 0.1Mb overlap. The windows reaching this threshold are then combined into regions, separate if there is no overlap otherwise one long region if there are overlaps. We think it is enough to specify the settings used and the function used so readers can themselves look for details in how the program runs the function.

L592: What do you consider as “top selection loci”?

R: We agree this is unclear. We meant the candidate regions under selection/XP-EHH regions that we present in more detail. We have now rephrased to:

L:722 “Canine TADs²⁶ from liver were also used to characterize XP-EHH regions that are presented in more detail in the main figures.”

L591: Explain TADs and what you did here.

R: We added the following to explain better:

L:723 “A TAD is described as a self-interacting genomic region where DNA sequences within a TAD physically interact with each other more often than with sequences outside the TAD. Thus, studying TADs may give indications of which genes are more likely to be regulated by the effect variants. The TADs can differ between tissues but the canine liver TADs can give us an estimation of what genes may be regulated in the nearby region to the variants defined in this study, even if the more appropriate tissue would have been skin.”

L595: What do you mean by “variants forming clusters of association”?

R: We realize this is confusing, we have now changed to “associated variants”:

L:731 “Associated variants from --assoc and/or --logistic models were regarded as potentially associated with AD.”

L607: Why did you use this distance?

R: 1Mb is the approximate size of a TAD, in which regulation between distant enhancers and genes usually occur.

This is described in ref: “Bonora et al., *A mechanistic link between gene regulation and genome architecture in mammalian development*, Curr Opin Genet Dev. 2014 Aug; 0: 92–101” 3C-based approaches have also identified self-associating chromatin domains of approximately 1Mb in size, TADs, that appear to be very stable across cell types and species, and are composed of complex networks of enhancer-promoter interactions that are restricted by the domains’ boundaries^{17,18}. These TADs appear to be the fundamental modular unit of chromatin organization.”

In addition to explaining TADs, we added a note about 1Mb from Bonora et al., 2014:

L:743 “To investigate more distant potential candidate genes in AD-associated loci, we extracted protein coding genes located within 1Mb (the approximate size of a TAD¹²⁸) from effect variants of each associated locus, denoted BayesR regions.”

L614: canine AD instead of CAD (because you use canine AD throughout the rest of the manuscript)?

R: Yes, we have now corrected this.

L616: Explain phyloP scores (e.g. score that describe evolutionary conservation)

R: We added the following sentence to

L:757 “The phyloP score is the log (P-value) under a null hypothesis of neutral evolution, and a positive score indicate evolutionary conservation where positions in the genome remain the same across many species because they are functional. In contrast, negative phyloP scores indicates accelerated evolution, corresponding to positive selection.”

Is there a difference between working/ show comparison and field/ no-field comparison? If not, stick to one terminology to avoid confusion.

R: Agreed. Referring to our previous discussion regarding working dogs of LRs, we have now decided to refer to gundogs vs. common type. This has been updated throughout the manuscript.

L115: Maybe better “(...) identifies fifteen risk loci for canine HD”?

R: Yes, changed.

L116: There are also dogs from other countries included according to Supp Table 2?

R: Germany, Netherlands, France, and Finland were included in the Swiss cohort. This is described in Materials and Methods under section “Sample collection and genotyping” but we clarified it:

L:617 “The Swiss cohort included samples from dogs collected in Switzerland, Netherlands, Finland, Germany, and France.”

We now also clarified this as a **Supplementary Table 1** footnote:

“*The samples from the Netherlands, Germany, France, and Finland were included in the Swiss cohort. “

Supplemental tables: use “.” as decimal separator instead of “,”

R: This is now corrected.

L134: Why was the loci on Chr 17 (with 9 effect variants) chosen and not Chr 36 (with 10 effect variants)?

R: We did further analyses using long-read sequencing (WGS) with the chr17 locus in mind (i.e., we chose individuals with the risk and control haplotypes for this locus) due to the overlap with the top human AD genetic association. This locus is of special interest due to the human-dog overlap of risk factors for AD and we performed detailed bioinformatics of this region specifically. Additional analysis based on the other loci presented in our study are also of interest but we chose to focus on the chr17 locus for that reason.

L145: Change to “overlapped with”

R: Done.

L156: Mention that this is the human reference genome

R: We now shortened that section down (to improve readability) and the 19 LD-variants are mentioned only briefly:

L:172 “A region of homozygosity (14kb) spans half of ACP6 and includes 19 variants in LD with chr17:d, out of which one variant overlapped both ATAC-seq, cCRE, and GeneHancer (Fig. 3d).”

Whereas the specifics about ATAC seq, cCREs and GeneHancers are defined in Materials and methods:

L:764: “The effect variants were intersected with BarkBase ATAC-seq data²⁶, and, for variants lifted to hg38, ENCODE cCREs²⁷⁻²⁹ (from UCSC Genome Browser) and GeneHancer³⁰ elements. LiftOver131 between genomes (canFam3.1, canFam4, and human hg38) was used to evaluate and compare functional and non-functional positions across assemblies.”

L185: how do you define “candidate region of selection”?

R: We have now clarified this definition in Materials and Methods and we specified the settings used by the function: (settings were threshold = 4, pval = TRUE, window_size = 1E6, overlap = 1E5, min_n_extr_mrk = 2) as a response by a previous comment.

We now also clarified in Results:

L:192 “In LR, GSD, and WHWT, a total of eight candidate regions of selection (XP-EHH regions) were identified. Regions were defined using a 1Mb window scan with 0.1Mb overlap and at least two variants with $-\log_{10}(p)$ XP-EHH above 4 (Fig. 1, Supplementary Fig. 2, and Table 2).”

L185: Which three breeds?

R: We rephrased, see above.

L195: It's "low scores for PC1"

R: Yes agreed, we changed to "low PC1 values".

Figure 4a: Differentiation between dog types is really hard to see in figure, consider using different symbols instead of point sizes?

R: We have changed **Fig 4a** and excluded the breed type symbols to clarify the picture, since combining both symbol shape and color made the whole figure unclear. We now also included color in the supplementary figures to discuss the breed types (as mentioned above). This applies also for **Fig 5a** (GSD) and **Supplementary fig. 4**.

L200: "allele C was more frequent in show subpopulation" where is this shown in Figure 4?

R: It is visualized in fig 4a. The sentences were rephrased to:

L:215 "The 115 selection variants on chromosome 3 were positioned across the *TBC1D1* gene and the top variant at chr3:74,218,744 (chr3:sel) was intronic to *TBC1D1* (**Fig. 4b**). The allele C of chr3:sel was more frequent in the common type subpopulation (**Fig. 4a**)."

L260: Please consider using "Genes under putative selection" instead of "selection genes" throughout the manuscript

R: Sounds good, this is changed.

L285: Rephrase this sentence, e.g. name the disease you found associations for.

R: L:309 "We defined 15 AD-associated loci using BayesR, represented by 54 effect variants across four dog breeds."

L286: What was the reason for choosing these cut-offs (not explained in Material and Methods)? Are there comparable values reported in other studies?

R: We explained this in previous responses, see above.

L293: According to this argument, would you expect "better" results for array data instead of WGS data? What was the motivation of using WGS data, considering that this goes along with certain bias (due to imputation) and you had to prune the data afterwards anyway due to computational restraints?

R: Imputed data can provide more accurate boundaries for candidate regions (better precision). The LD pruning excludes only the markers with extremely high LD. These markers do not add to the information as the markers are in high LD with other markers that remain in the dataset. Generally speaking, SNP arrays covers the genome fairly well but the imputed dataset increase the marker set approximately five times and this is even after LD pruning which shows that the additional markers add information (i.e., coverage). Yes, the absolute effect sizes are generally larger if fewer markers are included but you may also miss important regions when the coverage is lower. The exact effect size values are not really adding to the information/results as it is relative.

Conveniently, imputed data includes all markers in the region at the same time, which means we do not have to run the imputation to define candidate variants after the association study, i.e., we can go to the non-pruned imputed data to search for causative variants for each associated region.

We are addressing this issue in Discussion:

L:319 "A lower mean absolute effect size for each variant is also expected in a denser marker set since BayesR iterates the process of assigning variants to different effect size distributions and variants in high LD are randomly selected. On the other hand, the genomic positions indicated by effect variants are more precise and the risk of missing important regions associated with the trait is decreased in a denser marker set."

L296: Clarify that it is a risk loci for LRs

R: L:326 The AD-associated locus on chromosome 17, identified with BayesR in LR, overlaps with associated variants from two human AD meta-GWAS studies..."

L308 and following: Give gene names in italic

R: Gene names are written in italics but when referring to the protein, italics is not used.

L358: Do you have investigated the coat colour in your dogs (e.g. for clustering in PCA, for association with risk loci/ risk index)?

R: Now we also investigated coat color in our datasets. This is specified in responses to previous questions and added to the manuscript/supp info, for both LR and GSD.

L365: Clarify that it is a risk loci for GSDs

R: Yes, this need to be clarified, and it was in LR not GSD. The whole section about the chromosome 3 selection locus in LR in Discussion has been substantially revised see L:397-442.

L400-402: Please clarify what you mean here.

R: This refers to the following sentence: "Since allele T at chr19:sel was more frequent in controls of working type, the selection for a "work-desired trait" caused by variants affecting the *LRP1B* gene may have resulted in a pleiotropic action in immune cells (*LRP1B*) or a hitch-hiking effect in the skin (*KYNU*) by canine AD-protective variants."

We have now updated this sentence to:

R: L:466 "Since allele T at chr19:sel was more frequent in controls of working type, a possible scenario could be that specific variants affecting the *LRP1B* gene has influenced a "work-desired trait" in GSD that was selected for, and that additional (hitch-hiking) variants affecting (and potentially inhibiting) *KYNU* are AD-protective."

Pleiotropy and hitch-hiking were also mentioned in the introduction and here we merely refer to these effects as a possible action by the variants under selection. The description from introduction: "The hitchhiking effect is the unintended increase in allele frequencies of nearby variants at loci controlling another trait or disease. A pleiotropic effect can also be expected when genes responsible for the desirable trait also affect other phenotypes."

We believe it would be repetitive to describe these terms again.

L520: started with merged data set of all genotyped dogs, why different genotype number for breeds in Supp Table1? I assume you applied the QC within breeds?

R: Yes, we started with merged dataset and imputed the whole set. The data was then split up by breed and QC was performed. In parallel, we used the SNP-chip genotypes to run relatedness filtering and PC calculations. We have clarified the QC-steps in the text now:

In Materials & Methods:

L:627 "The CanineHD 230K BeadChip is an extension of the CanineHD 170K BeadChip and we started with a merged genotype dataset consisting of 167,211 SNPs and 1,152 dogs from four breeds."

L:632 "Quality control (QC) was performed per dog breed (plink --geno 0.05 --mind 0.05 --maf 0.05)."

Before Imputation:

L:653 "Quality parameters used in plink prior to imputation were --maf 0.0001, --geno 0.05, and --mind 0.05 and the dataset before imputation consisted of 1,152 dogs (347 LR, 294 GR, 231 GSD, and 280 WHWT) and 148,889 SNPs."

After Imputation:

L:669 "After imputation, the dataset was split into each breed and analyzed breed-wise (QC: plink --geno 0.02 --mind 0.05 --maf 0.05)."

REVIEWERS' COMMENTS:

Reviewer #1 (Remarks to the Author):

Dear Editor and Authors,

I concerned current manuscript is OK for publication after proper response and revision based on all the reviewers's comments.

With the kindest regards,

Guo-Dong

Reviewer #2 (Remarks to the Author):

The authors' revisions have solved all my concerns, and the article is of great importance not only for canine genetic disease studies but also providing clues for human atopic dermatitis research. They performed a large amount of data analysis from different aspects. The conclusion is quite convincing and the research will have influence on canine inherited disorder community. I think this paper can be accepted.

Reviewer #3 (Remarks to the Author):

Dear authors,

thanks for addressing suggested changes and potential issues in this thorough revision. The findings as reported here are of great interest and the clarifications/ additional details strenghten the conclusions/ reproducibility of the analyses. I recommend the manuscript for publication.

Congratulations on your research!